# PhotoAgent: Exploratory Visual Aesthetic Planning with Large Vision Models

**Mingde Yao** [1 2]  **Zhiyuan You** [1]  **King-Man Tam** [3]  **Menglu Wang** [4]  **Tianfan Xue** [1 2 5]

## Abstract

With the recent fast development of generative models, instruction-based image editing has shown great potential in generating high-quality images. However, the quality of editing highly depends on carefully designed instructions, placing the burden of task decomposition and sequencing entirely on the user. To achieve autonomous image editing, we present **PhotoAgent**, a system that advances image editing through explicit aesthetic planning. Specifically, PhotoAgent formulates autonomous image editing as a long-horizon decision-making problem. It reasons over user aesthetic intent, plans multi-step editing actions via tree search, and iteratively refines results through closed-loop execution with memory and visual feedback, without requiring step-by-step user prompts. To support reliable evaluation in real-world scenarios, we introduce UGC-Edit, an aesthetic evaluation benchmark consisting of 7,000 photos and a learned aesthetic reward model. We also construct a test set containing 1,017 photos to systematically assess autonomous photo editing performance. Extensive experiments demonstrate that PhotoAgent significantly outperforms existing methods in both instruction faithfulness and visual quality across a diverse range of editing scenarios. The project page is https://mdyao.github.io/PhotoAgent/.

## 1. Introduction

Recent instruction-based image editing models (Instruct-Pix2Pix (Brooks et al., 2023), SDXL (Podell et al., 2023), SD (Rombach et al., 2022), GPT-4o (OpenAI, 2024a), Flux.1 kontext (Labs et al., 2025), Bagel (Deng et al., 2025), etc.) enable amateur users to achieve professional photo edits through natural language commands (*e.g.*, remove the passersby), rather than solely manipulating low-level sliders (*e.g.*, brightness and color). This shift broadens the scope of computational photography, moving beyond fidelity to the **captured scene** toward fidelity to the user's **aesthetic intent**, thereby democratizing powerful photographic expression (Wang et al., 2024a; Liu et al., 2025).

Despite these advances, a critical bottleneck remains: these powerful models fundamentally rely on continuous user involvement, as shown in Fig. 1. Their effectiveness largely depends on the user's ability to design precise and sequential instructions, which is difficult for amateur users. This reliance introduces several fundamental limitations: (1) **Expertise barrier**: Effective interaction requires expert knowledge. Amateur users often struggle either with designing articulate and precise editing instructions (*e.g.*, decomposing "make my photo better" into detailed steps) or with evaluating whether editing results meet professional quality standards. (2) **Algorithm selection**: Different editing tasks require different specialized models. A single model may not be sufficient for all tasks, so users need to switch between models to achieve the desired results. (3) **Interaction complexity**: These models often require users, even professional ones, to issue multiple iterative commands, which is inherently time-consuming and prevents full automation for batch processing.

We argue that the next frontier in computational photography is not merely a single powerful editor or processor (Brooks et al., 2023) (Hertz et al., 2022) (Wang et al., 2024a) (Liu et al., 2025), but an autonomous editing agent that can enhance photos without requiring expert-level operation. Such an agent would emulate the decision-making process of a human photo editor, who strategically selects and sequences tools based on an assessment of the image's needs, and edits with specific tools. Recently, large vision and multimodal models (LVMs) (Deng et al., 2025) (Liu et al., 2023) (Brooks et al., 2023) (Wang et al., 2024a) have demonstrated remarkable perception and instruction-conditioned editing capabilities, making an autonomous editing agent feasible.

In this paper, we introduce **PhotoAgent**, a novel autonomous system that integrates large vision and multimodal models (LVMs) with a suite of editing tools into a

[1]MMLab, CUHK [2]CPII under InnoHK [3]Institute of Science Tokyo [4]Univerisity of Science and technology of China [5]Shanghai AI Lab. Correspondence to: Tianfan Xue <tfxue@ie.cuhk.edu.hk>.

*Proceedings of the 43rd International Conference on Machine Learning*, Seoul, South Korea. PMLR 306, 2026. Copyright 2026 by the author(s).

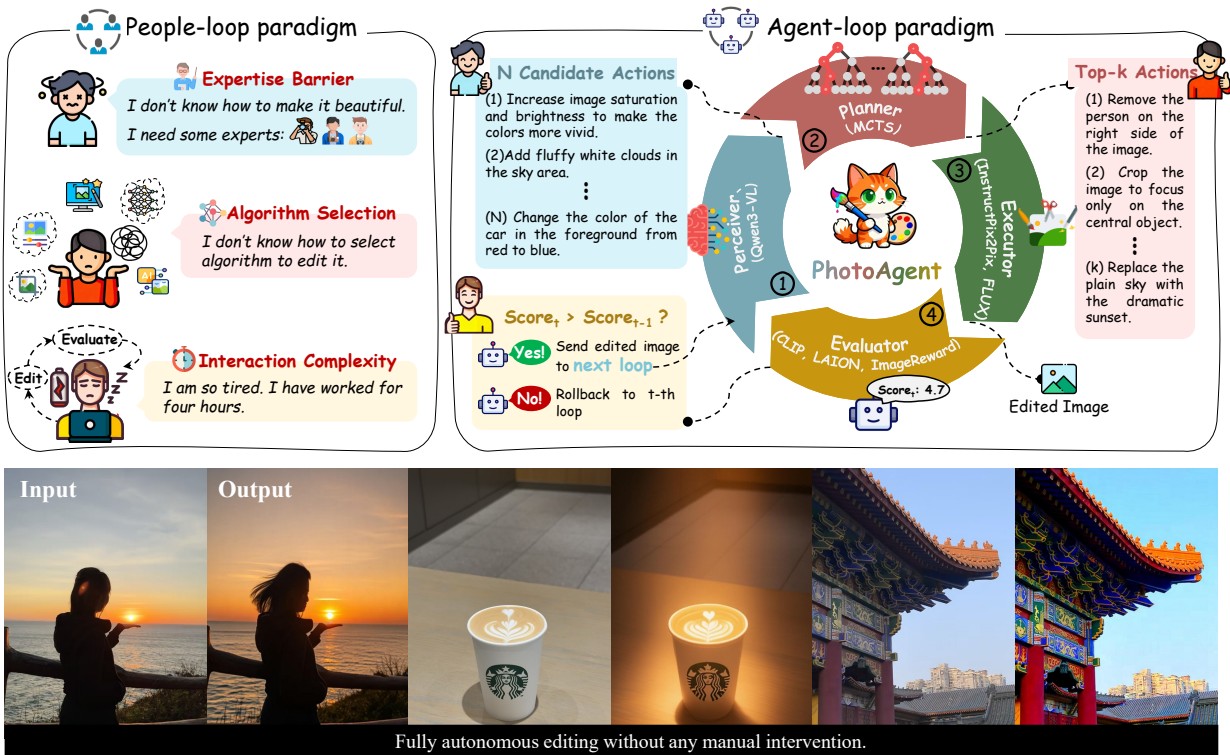

*Figure 1.* PhotoAgent autonomously performs high-level, semantically meaningful edits aligned with human aesthetic, moving beyond low-level color, contrast, or illumination tweaks. **Upper-Left**: Human-loop, where users iteratively inspect the image, propose edits, and apply changes until satisfied. **Upper-Right**: PhotoAgent, where the process runs autonomously. **Bottom**: Edited photos. Note that PhotoAgent also supports user-guided editing (Fig. 6).

coherent framework, enabling fully automated, high-quality photo editing. As illustrated in Fig. 1, PhotoAgent introduces **exploratory visual aesthetic planning** within a **closed-loop** framework. Unlike open-loop systems (*e.g.*, GenArtist (Wang et al., 2024b)) that execute linear action sequences without feedback, PhotoAgent continuously evaluates its edits and strategically explores the editing space. This helps to avoid both short-sighted decisions and irrecoverable artifacts that commonly happen in greedy approaches, enabling coherent and high-quality results. In addition, PhotoAgent enables **context editing**, moving beyond the low-level adjustments (*e.g.*, color, contrast, illumination) that existing photo-editing agents primarily perform (Lin et al., 2025; Dutt et al., 2025; Zuo et al., 2025). This is achieved through programmatic control over a rich library of editing actions and flexible editing tools, enabling semantically meaningful manipulations such as *adding a sun to a dim sky*, *making the scene feel more vibrant and lively*, or *modifying objects in the scene*.

To achieve this, PhotoAgent consists of four core components: a perceiver, a planner, an executor, and an evaluator. The process begins with a VLM-based **perceiver** (*e.g.*, Qwen3-VL (Bai et al., 2025a)) that interprets the input image and produces a set of semantically meaningful editing actions. These candidate actions are then passed to a Monte

Carlo Tree Search (MCTS)-based **planner** (Chaslot et al., 2008) (Browne et al., 2012), which explores possible editing trajectories in a tree structure and selects the top-K most promising actions. This exploratory mechanism ensures that our system embodies exploratory visual aesthetic planning, avoiding myopic decisions. The selected actions are subsequently **executed** using either advanced image generation tools (*e.g.*, Flux.1 Kontext (Labs et al., 2025)) or traditional image processing libraries (*e.g.*, OpenCV/PIL (Bradski, 2000)). Finally, the **evaluator** integrates feedback from multiple scoring modules, allowing only those actions that positively contribute to the image's aesthetic quality to pass. By iterating through this **perceive–plan–execute–evaluate** cycle, PhotoAgent forms a fully closed-loop process, enabling autonomous and reliable progress toward the final editing goal.

Additionally, one major challenge in this design is that existing image quality evaluation methods are insufficient for user-driven photo editing, also referred to as user-generated content (UGC). The core issue lies in the composition of existing datasets, where existing datasets are overly generic, containing AI-generated images, screenshots, advertisements, and posters, rather than authentic user-captured photographs. To address this, we introduce **UGC-Edit**, a dataset of 7,000 real user photos annotated with human aesthetic

scores. We also train a reward model on UGC-Edit, enabling reliable evaluation of aesthetic quality for multi-step image editing. Finally, to comprehensively evaluate the editing, we construct a test set of real photographs consisting of 1,017 images, on which our system achieves state-of-the-art results across quantitative metrics, qualitative assessment, and user studies.

In summary, this work makes the following contributions:

- We propose PhotoAgent, an autonomous editing system that integrates a closed-loop architecture with a suite of editing and evaluation tools, enabling robust multi-step editing.

- We introduce a visual aesthetic planner to explore sequences of editing actions over long horizons, enabling deliberate, goal-driven image editing.

- We present the UGC-Edit dataset and introduce a reward model to support aesthetic research in autonomous image editing. We also introduce a test set of real photographs for evaluating autonomous photo editing.

- Extensive experiments demonstrate that our complete system achieves significant improvements in editing quality.

**Conflict of Interest Disclosure**. The authors declare no financial conflicts of interest.

## 2. Related Work

**Image Editing** Early pioneering works primarily leverage Generative Adversarial Networks (GANs) (Goodfellow et al., 2014) or conditional encoder-decoder architectures for tasks like style transfer and attribute manipulation. For example, CycleGAN (Zhu et al., 2017) proposes unpaired image-to-image (I2I) translation, and StarGAN (Choi et al., 2018) enables multi-attribute manipulation within a single model. However, these approaches are inherently limited since their editing capabilities are confined to the narrow distribution of their training data, which often struggle with open-vocabulary requests. They frequently produce low-resolution or artifact-ridden outputs.

A paradigm shift was ushered in by the advent of powerful diffusion models (OpenAI, 2024b; Wu et al., 2025) and their integration with natural language. Models like Stable Diffusion (AI, 2024) treat image editing as conditional image generation, where the input image serves as a foundational condition. Recent methods (*e.g.*, Prompt-to-Prompt (Hertz et al., 2022), InstructPix2Pix (Brooks et al., 2023)) manipulate the features in latent space to enable highly flexible editing following open-vocabulary instructions. This

progress continues with next-generation architectures based on flow matching (*e.g.*, Flux (Labs et al., 2025)) and the integration of powerful Multimodal Large Language Models (MLLMs) like GPT-4o (OpenAI, 2024a), Show-o (Xie et al., 2024), Bagel (Deng et al., 2025), Nano Banana (Google, 2025) and HunyuanImage-3.0 (Cao et al., 2025), which aim to tightly couple reasoning and generation. Despite these remarkable advances, a critical limitation exists. These models act primarily as single-step, static executors. Their performance is highly sensitive to meticulously engineered, low-level prompts, placing the burden of designing instructions and evaluations on the amateur user. These limitations prevent the method from handling complex, autonomous multi-step editing tasks, highlighting the need for a higher-level, planning-based framework.

**Planning with Autonomous Agents** To overcome the above limitations, a promising direction is to design an autonomous agent framework capable of multi-step planning and execution. Early works such as AlphaGo (Silver et al., 2016) employ planning algorithms like Monte Carlo Tree Search (MCTS) to navigate state spaces. Recently, LLM-based agents leverage LLM's reasoning capability to decompose tasks into sequences of actions (*e.g.*, HuggingGPT (Shen et al., 2023), ReAct (Yao et al., 2023), and Voyager (Wang et al., 2023)).

Within computer vision, works have explored integrating planning into image editing tasks. Some approaches, such as JarvisArt (Liu et al., 2025) and MonetGPT (Dutt et al., 2025), leverage an LLM as a planner to parse a complex instruction into a sequence of calls to specialized image processing software. However, existing methods mainly focus on low-level editing tasks, such as color, tone, or exposure adjustments using procedural software tools like Lightroom or GIMP, which are limited to pure retouching.

More recent researches begin to explore directly applying MCTS and other search strategies to the text-to-image (T2I) generation process itself, building a search tree in the latent or textual space to find sequences of actions that better satisfy a high-level goal (Shi et al., 2025). However, existing methods (Lin et al., 2025; Zuo et al., 2025; Dutt et al., 2025) have no planning capability for instruction-based image editing. Our approach addresses this through an MCTS planner that considers internal simulation with external execution. We also employ a learned reward model trained on user preferences to guide the search. This combination enables robust planning with a diverse toolset and is supported by a new editing-specific benchmark for evaluation.

**Image Evaluation** In an automated image editing pipeline, the evaluator is important, as it defines the reward function that guides the agent's actions and determines the final output. Traditional full-reference image quality metrics, such as Peak Signal-to-Noise Ratio (PSNR) and Structural Simi-

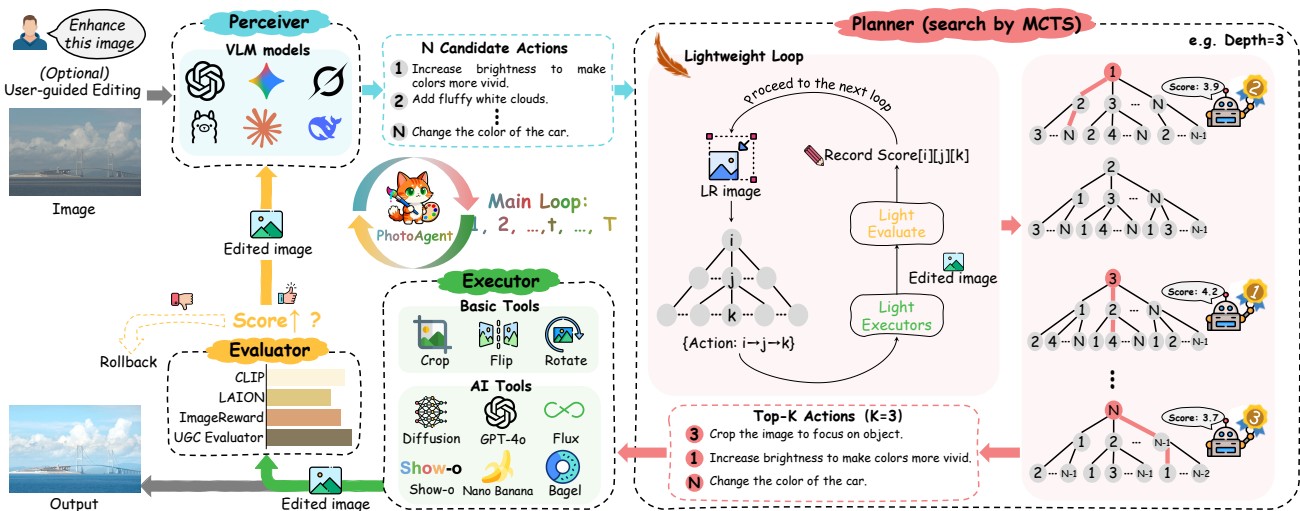

*Figure 2.* Detailed loop of PhotoAgent. First, **Perceiver** extracts semantic cues from the current image and proposes $N$ candidate editing actions. Second, **Planner** explores the candidate actions through iterative rollouts, scoring, and pruning to progressively refine edits and select the action that achieves the optimal result. Then, the **executor** applies these edits while the **evaluator** scores intermediate results, invoking re-planning when the score is unsatisfactory.

larity Index (SSIM) (Wang et al., 2004), are not suited for this open-world setting. They require a ground-truth target image, which is obviously impossible for creative editing tasks.

The community then turns to no-reference metrics, including distribution-based measures like Fréchet Inception Distance (FID) (Heusel et al., 2017), aesthetic predictors (discus0434, 2024), or CLIP-based image-text alignment scores (Radford et al., 2021). While a step forward, these metrics are often too broad to provide reliable, fine-grained signals for specific editing tasks on user-generated content (UGC). They cannot capture the subtle quality differences that are crucial in specific image editing tasks, such as aesthetic-oriented editing. To address this limitation, we introduce a specialized UGC evaluation dataset and train a reward model on the dataset. The reward model is adopted from a pretrained vision-language model (VLM) that contains inherent knowledge. The dataset and model enable learning of aesthetic evaluation, providing precise feedback to guide the agent toward high-quality results.

## 3. PhotoAgent

We propose PhotoAgent, an autonomous image editing system capable of executing multi-step editing tasks through a structured, closed-loop framework. As shown in Fig. 1, the system comprises four core components: a perceiver that interprets the input image and generates candidate editing actions, an MCTS-based planner that explores and selects potential editing actions, an executor that applies the edits, and an evaluator that assesses the editing results. In addition, PhotoAgent incorporates a tool-selection module to dynam-

ically choose suitable editing tools and a memory module that records editing history, enabling informed planning, reflection, and consistent decision-making across multiple editing steps. The system operates iteratively through a perceive–plan–execute–evaluate cycle, where the closed-loop process continues until the editing objective is met or a termination condition is satisfied.

**Perceiver: Instruction Generation** The perceiver utilizes a VLM, *e.g.*, LLaVA (Liu et al., 2023), Qwen3-VL (Bai et al., 2025b;a), to analyze the visual input $I_t$ and generate a set of $K$ diverse and atomic editing actions $\{a_t^k\}_{k=1}^K$. To this end, we introduce a structured, context-aware multimodal prompting scheme that conditions the VLM on both the current visual scene and aesthetic attributes, enabling the perceiver to act as an aesthetic-driven instruction generator. This structured, context-aware scheme has the following capabilities.

**(a).** The perceiver supports both **fully autonomous editing**, where no explicit user command is provided, and **user-guided editing**, where users may express intent through mood, atmosphere, or feeling rather than concrete objects or operations. **(b).** We also let the perceiver infer the scene type and use it with the **scene-aware** prompt. This allows the agent to generate tailored strategies for different types of scenes. For example, for images with a human subject, we prioritize maintaining the character's appearance while allowing more aggressive modifications to the background. **(c).** Moreover, we perform **memory** mechanisms that record the outcomes of each editing round to guide subsequent instruction generation. Over time, these records form a static strategy memory that helps the agent improve continuously

and produce diverse, contextually appropriate edits. See Appendix H for detailed prompt and memory (history) design.

**Planner: MCTS-Based Action Exploration** The planner chooses the candidate actions through an MCTS-based planning process, as shown in Fig. 2. Specifically, unlike existing methods (Lin et al., 2025; Dutt et al., 2025; Zuo et al., 2025) that edit without planning, our planner enables the agent to simulate sequences of future edits, evaluating their long-term consequences before execution. This approach avoids short-sighted decisions and irreversible mistakes. To achieve this, MCTS consists of four phases: selection, expansion, simulation, and backpropagation.

Formally, each **state** $s_k$ denotes the current image after $k$ editing steps. The **action space** consists of the $K$ candidate editing instructions proposed by the Perceiver, which are treated as discrete choices by the MCTS planner. Instructions are represented in natural language, as they are directly consumed by downstream editing executors.

In the *selection* phase, the planner starts from the root node representing the current image and chooses which candidate edits to explore next. These candidates come from the perceiver's output. The traversal balances exploration of new edits with exploitation of high-reward ones. When reaching a leaf node, the *expansion* phase adds new child nodes representing potential editing actions. For example, when evaluating an action like "adjust the color balance to enhance the blue of the sky and the green of the water", it creates a new node to represent the resulting image state.

In the *simulation* phase, we evaluate candidate edits efficiently using a fast-approximation environment. To speed up simulations, we use reduced-resolution processing, which preserves essential visual and semantic information. We verify that this approximation does not introduce a significant sim-to-real gap, as demonstrated in Appendix A.

Finally, during *backpropagation*, we calculate the reward value and propagate it back through the tree. This updates the visit count and average reward of each visited node, helping the selection phase make better decisions. After a number of simulations, the algorithm selects the action with the highest average reward or the most visits from the root node for actual execution.

**Executor: Action Execution** Then, the executors actually run the selected actions on the image. In practice, we select the top-K actions rather than only the action with the highest score, which ensures robustness and avoids simulation inaccuracies in the previous step. For each action, our system selects between traditional operators, *e.g.*, color adjustment or cropping via OpenCV/PIL (Bradski, 2000)), and advanced generative models, *e.g.*, FLUX.1 Kontext (Labs et al., 2025) or Step1X-Edit (Liu et al., 2025). We then employ the evaluator to evaluate all results, retaining only the highest-scoring

output as the next state $I_{t+1}$. This approach ensures that our final decisions are grounded in real outcomes rather than simulated estimates, significantly improving the reliability of our editing trajectory.

**Evaluator: Outcome Evaluation** The evaluator assesses the set of edited images $\{I_t^k\}_{k=1}^K$ produced by executing the top-K actions, and outputs each an assessment score $\{r_t^k\}_{k=1}^K$. PhotoAgent employs an ensemble evaluation strategy that integrates traditional no-reference metrics (such as NIQE (Mittal et al., 2012b) and BRISQUE (Mittal et al., 2012a)), modern instruction-based assessment (such as CLIP-based aesthetic scoring (Radford et al., 2021; Schuhmann et al., 2022) and instruction-following evaluation (Liu et al., 2023)), and customizable perceptual models (see Section 4), to provide a comprehensive evaluation. We provide detailed settings in the Appendix C.

The highest candidate score is compared against the score of the input image $I_{t-1}$. If an improvement is observed, the corresponding image is selected as the next state. Otherwise, the system reverts to $I_{t-1}$. The process terminates when the maximum number of steps is reached or updates no longer change the result.

## 4. Evaluation for Editing Systems

Effective evaluation is critical for autonomous image editing, where systems must assess aesthetic quality in a way that aligns with human preferences and directly guides multi-step decision-making. Existing image quality metrics are largely designed for generic images and are ill-suited for user-generated photos, especially in editing scenarios that require fine-grained and subjective judgments. To address this limitation, as shown in Fig. 3, we introduce a comprehensive evaluation framework that includes (i) a UGC-specific preference dataset for training aesthetic reward models, (ii) a learned reward model tailored to user-generated photos. Furthermore, we construct a real-world photo editing benchmark for evaluating end-to-end editing performance.

**UGC-Edit Dataset** As shown in Fig. 3, we introduce **UGC-Edit**, a dataset of approximately **7,000** authentic user-generated photos designed for training aesthetic evaluation models in photo editing systems. Images are sourced from the LAION Aesthetic dataset (Schuhmann et al., 2022) and the RealQA benchmark (Li et al., 2025). As they contain diverse web images beyond real user photos, we apply a two-stage filtering process: a vision-language model first categorizes image types, followed by manual verification to retain only images with clear UGC characteristics. We merge the two sources and normalize all aesthetic scores to a unified 1–5 scale. This dataset serves exclusively as supervision for training a UGC-specific reward model aligned with human aesthetic preferences.

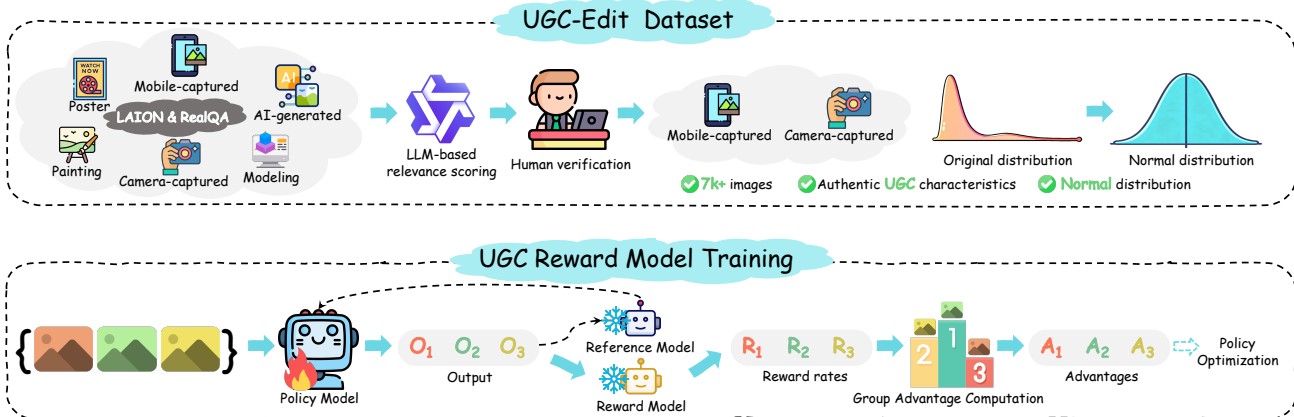

*Figure 3.* Pipeline for constructing the UGC-Edit Dataset and training reward model. We start with a diverse pool of source images from LAION (Schuhmann et al., 2022) and RealQA (Li et al., 2025). Each image is processed through a structured prompt with Qwen3-VL (Wu et al., 2025) for UGC classification. The images are then filtered by human annotators. Finally, a reward model is trained via GRPO (Shao et al., 2024) to predict fine-grained quality scores.

Unlike RealEdit (Sushko et al., 2025), which targets editing model training, and MultiRef (Chen et al., 2025), which focuses on controllable generation, UGC-Edit is designed as a scoring dataset to train an aesthetic reward model, enabling closed-loop feedback in agent-based editing.

**UGC Reward Model** We train a UGC-specific reward model on UGC-Edit to predict fine-grained aesthetic scores reflecting human preferences, as shown in Fig. 3. The model is initialized from a pretrained vision-language model (Qwen2.5-VL (Bai et al., 2025b)) and optimized using Group Relative Policy Optimization (GRPO) (Shao et al., 2024), which learns from relative rankings within image groups. This training strategy improves robustness to annotation noise and enables the model to capture subtle aesthetic cues for guiding multi-step photo editing. Importantly, the learned reward model constitutes one component of our evaluation framework. Final performance is assessed using a comprehensive evaluation protocol that combines multiple complementary metrics and human judgments. Detailed analyses are provided in Appendix B and C.

**Editing Benchmark for Final Evaluation** To evaluate the end-to-end performance of photo editing systems, we construct a separate benchmark consisting of 1,017 real-world photographs captured by different users and devices. It covers a diverse range of common photographic scenes, including portrait photography, natural landscapes, urban and architectural scenes, food photography, everyday objects, and low-light or night-time imagery. Each image is edited using multiple baseline methods for final evaluation. We report quantitative metrics, qualitative comparisons, and user study results on this benchmark to assess real-world editing effectiveness.

## 5. Experiments

### 5.1. Implementation Details

We choose two groups of baselines for comparison, including non-agent methods and agent methods. For non-agent methods, we compare with InstructPix2Pix (Brooks et al., 2023), SDXL+Prompt (Podell et al., 2023), and Flux.1 Kontext (Labs et al., 2025), which performs editing in a single step without planning capabilities. We use the vague editing prompt (*i.e.*, "make this image better") to reflect realistic scenarios with ambiguous user intent. For agent methods, we compare with HuggingGPT (Shen et al., 2023) (which generates all editing commands in a single call), ReAct (Yao et al., 2023) (Open-loop, iteratively plans and executes without evaluation), and ReAct (Yao et al., 2023) (Closed-loop, iteratively plans and incorporates an evaluator to decide action retention).

We calculate two types of metrics: semantic alignment and non-reference image quality. For semantic alignment, we use CLIP Similarity (↑) (Radford et al., 2021) to measure how well the edited image preserves the original content. For image quality, we report ImageReward[1] (↑) (Xu et al., 2023) to approximate human preference alignment, BRISQUE (↓) (Mittal et al., 2012a) for non-reference image quality, and Laion-Reward (↑) (Schuhmann et al., 2022) for general aesthetic preference. Additionally, we report UGC Score (↑), which is calculated from our reward model fine-tuned on user-generated content (Section 4) to better reflect users' aesthetic preferences.

---

[1]We use the implementation of https://github.com/RE-N-Y/imscore.

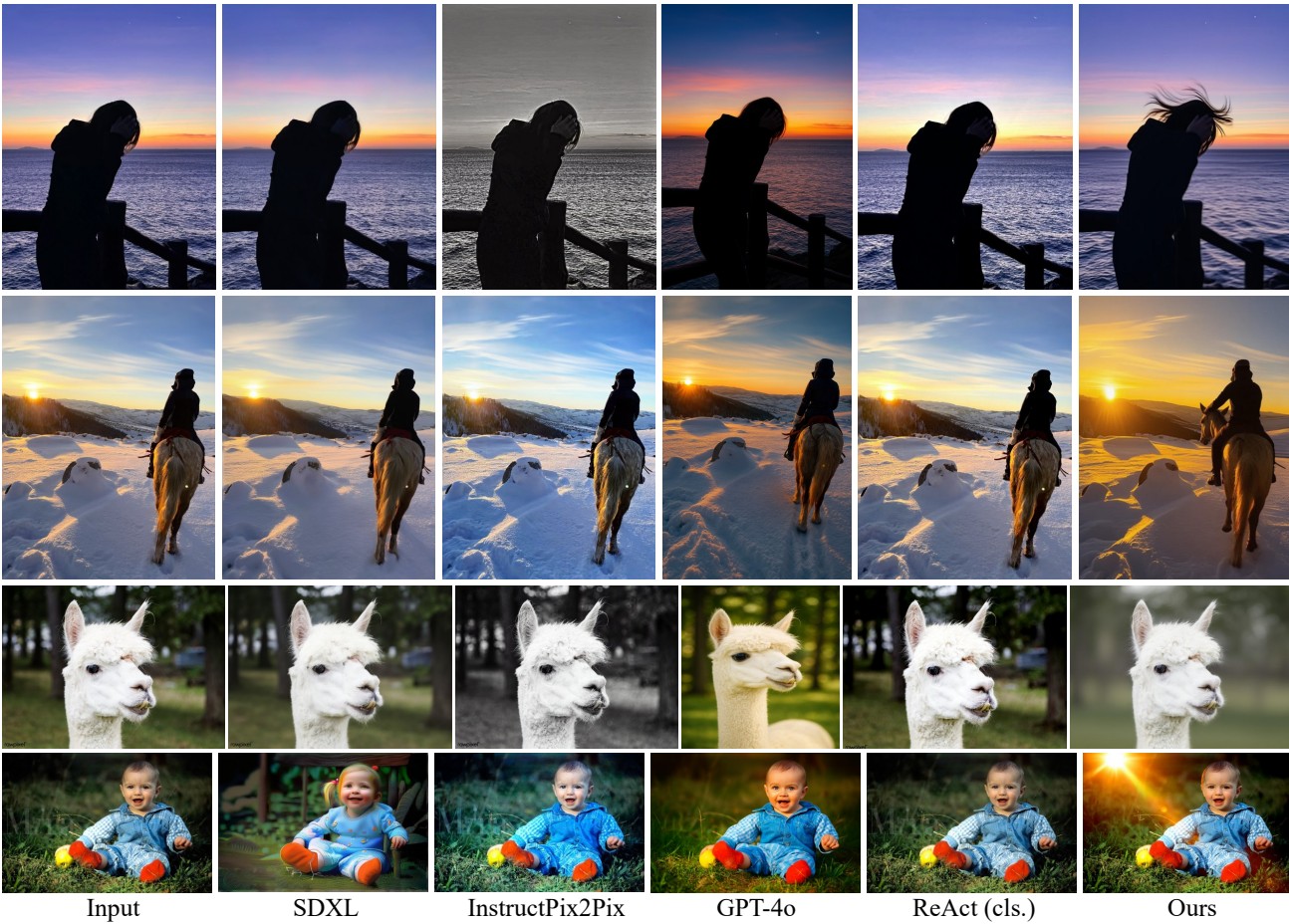

|       |      |              |        |             |      |
|-------|------|--------------|--------|-------------|------|
| Input | SDXL | InstructPix2Pix | GPT-4o | ReAct (cls.) | Ours |

*Figure 4.* Qualitative results. PhotoAgent generates visually pleasing edits by autonomously improving color harmony, composition, and aesthetic expressiveness, often introducing a stronger sense of visual dynamics and atmosphere. Baseline methods tend to produce incomplete or less coherent outputs.

## 5.2. Main Results

**Quantitative Results** We show the quantitative results in Table 1. It can be seen that PhotoAgent achieves the best BRISQUE score, which reveals the effectiveness of our framework. In contrast, GPT-4o exhibits an over-editing intent, often outputting overly vivid colors and exaggerated contrast. While such outputs may score well on perceptual metrics (e.g., UGC Score), they can introduce significant image distortions, as also reflected in their degraded BRISQUE and CLIP Similarity scores. PhotoAgent is the only method that ranks in the top-2 across *all five* metrics simultaneously, demonstrating consistent and balanced superiority over methods that excel on only a subset. In addition, our method attains competitive performance on other metrics such as ImageReward. As for agent-based baselines, their overall performance is limited. In open-loop settings, this is mainly because they lack visual feedback, which can cause errors to accumulate and the system to drift away from the correct trajectory. In closed-loop settings, their performance is also constrained, which may lead to subop-

timal or short-sighted decisions. The results demonstrate the effectiveness of the PhotoAgent, especially in producing consistent improvements in both semantic alignment and aesthetic quality.

**Qualitative Results** We also show qualitative comparisons in Fig. 4 to clearly demonstrate PhotoAgent's effectiveness. We observe that non-agent methods, such as GPT-4o, often apply generic edits and fail to address specific issues when given vague instructions (*e.g.*, "make this image better"). Meanwhile, we find that agent-based baselines (Shen et al., 2023; Yao et al., 2023), often suffer from error accumulation and make short-sighted planning decisions, resulting in unsatisfactory visual output. In contrast, PhotoAgent effectively explores multiple editing paths through a closed-loop planning mechanism, and progressively selects and executes the most appropriate editing actions.

**Comparison with Advanced Single-Step Executors.** To disentangle the contribution of planning from executor choice, we further compare against Grok-Imagine and Nano-

*Table 1.* Quantitative comparison of different planning strategies. The best results are in **bold**, and the second best are underlined.

| Methods | CLIP Similarity ↑ | ImageReward↑ | BRISQUE↓ | Laion-Reward ↑ | UGC Score↑ |
|---|---|---|---|---|---|
| GPT-4o (OpenAI, 2024a) | 0.6015 | **0.4115** | 0.7215 | 0.5131 | **4.210** |
| InstructPix2Pix (Brooks et al., 2023) | 0.6123 | 0.3824 | 0.6976 | 0.4826 | 3.428 |
| SDXL (Podell et al., 2023) | 0.6079 | 0.3801 | 0.7189 | 0.4944 | 3.277 |
| Flux.1 Kontext (Labs et al., 2025) | 0.6037 | 0.3971 | 0.6831 | 0.4973 | 3.561 |
| HuggingGPT (Shen et al., 2023) | 0.6006 | 0.3993 | 0.6992 | 0.4921 | 3.420 |
| ReAct (Open-loop) (Yao et al., 2023) | 0.6059 | 0.3872 | 0.6485 | 0.4989 | 3.175 |
| ReAct (Closed-loop) (Yao et al., 2023) | 0.6027 | 0.3962 | 0.6422 | 0.5011 | 3.258 |
| PhotoAgent (Ours) | **0.6254** | 0.4079 | **0.6217** | **0.5134** | 4.176 |

*Table 2.* User study results: percentage of votes selecting each method as the best.

| HuggingGPT | ReAct (cls.) | GPT-4o | PhotoAgent (Ours) |
|---|---|---|---|
| 12.6% | 15.2% | 30.2% | 42.0% |

Banana2, which are among the most capable closed-source single-step image editing models and might also serve as executor components within our system. As illustrated in Appendix D, while these models produce visually reasonable outputs, they exhibit characteristic failure modes under ambiguous instructions: Grok-Imagine tends to over-enhance, introducing exaggerated or dreamlike effects (e.g., artificial lens flares and oversaturated skies), while Nano-Banana2 applies edits without clear aesthetic guidance, sometimes altering the main subject unnecessarily. In contrast, PhotoAgent iteratively explores and refines editing trajectories, preserving original content when the image is already visually appealing and producing more coherent aesthetic enhancements otherwise. These qualitative results confirm that the performance gains of PhotoAgent arise from the planning framework itself, rather than from the choice of executor.

**User Study.** We conduct a user study with 20 participants across 26 real-world editing scenarios, collecting 540 preference votes. Participants with diverse photography and editing experience were asked to select the most aesthetically pleasing result while preserving the original image content. All outputs were presented in randomized and unlabeled order under a forced-choice protocol. As shown in Table 2, PhotoAgent is consistently preferred over all baselines, demonstrating its effectiveness in real-world settings.

## 5.3. Ablation Studies

We perform ablation studies to verify the key designs of PhotoAgent. We examine the effect of the UGC evaluator by removing it from the framework, which significantly decreases aesthetic metrics like Laion-Reward. This confirms the reward model's effectiveness in editing preferences. In

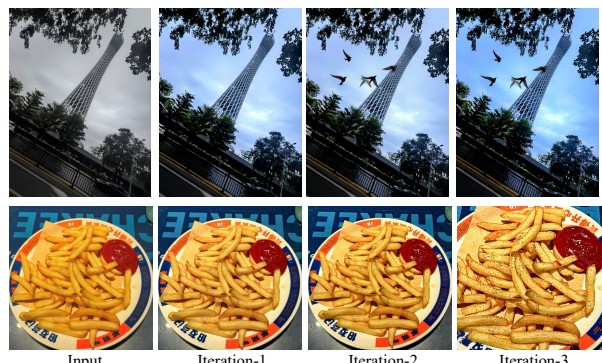

| Input | Iteration-1 | Iteration-2 | Iteration-3 |

*Figure 5.* The editing process of our PhotoAgent over three iterations.

addition, we investigate the importance of simulation times. Limiting the number of MCTS simulations to 10 results in suboptimal decisions, which demonstrates the necessity of strategic planning. Likewise, reducing the MCTS search depth to 1 can effectively implement greedy selection, leading to lower performance on multi-step edits. Overall, our results indicate that the evaluator, the depth of planning, and the number of simulations all play important roles in PhotoAgent's performance. Additional details are provided in the Appendix.

## 5.4. Analysis

**Editing Process** As shown in Fig. 5, PhotoAgent first adjusts the overall tone to significantly enhance the image's aesthetic quality. Based on it, PhotoAgent can further edit specific objects, such as flying birds, which makes scene appear more lively and dynamic. This iterative strategy allows PhotoAgent to simultaneously preserve the original content and improve aesthetics, highlighting its effectiveness and robustness in producing visually compelling results.

**User-guided Editing** PhotoAgent supports not only fully autonomous editing, but also user-guided editing, which is common in real-world usage. In the user-guided setting, users are not required to specify concrete objects or explicit

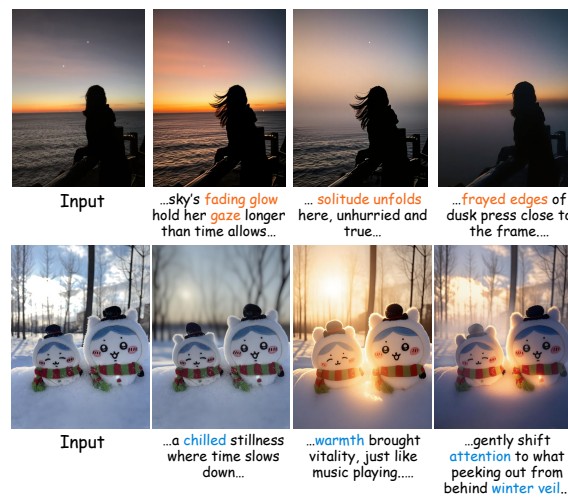

*Figure 6.* PhotoAgent with user-guided prompts.

editing operations. Instead, they may express high-level intent through abstract descriptions such as mood, atmosphere, or emotional tone. As illustrated in Fig. 6, PhotoAgent is able to interpret such guidance and produce visually appealing results with distinct styles under different prompts. This flexibility enables PhotoAgent to effectively align with the real-world application scenarios.

**Terminate Condition** PhotoAgent employs two complementary early stopping strategies to prevent unnecessary edits on high-quality images: (1) a maximum iteration limit, which forces termination after a predefined number of steps, and (2) a no-improvement criterion, which stops editing if the evaluator detects no significant score improvement over N consecutive iterations. These mechanisms ensure that high-quality images are not over-edited.

## 6. Conclusion

We present PhotoAgent, an autonomous image editing system that reframes photo editing as a sequential decision-making process, reducing the reliance on precise human instructions. The novelty arises from the coordinated interaction of multiple modules rather than innovating on the underlying editing models themselves. Specifically, the system integrates four key components: an LLM-based perceiver, an MCTS-driven exploration strategy, a tool-based executor, and a VLM-based evaluator, forming a closed-loop framework supported by the proposed UGC-Edit dataset. Experimental results demonstrate that PhotoAgent outperforms existing methods in producing semantically coherent and aesthetically consistent enhancements. The framework provides a foundation for more photo editing in future work.

## Acknowledgements

The work is supported by the National Key R&D Program of China (No. 2025YFE0201300). This study is supported in part by the Centre for Perceptual and Interactive Intelligence (CPII) Ltd., a CUHK-led InnoCentre under the InnoHK initiative of the Innovation and Technology Commission of the Hong Kong Special Administrative Region Government.

## Impact Statement

This work presents PhotoAgent, an autonomous image editing system that lowers the barrier to high-quality photo editing and enables broader access to professional-level enhancement. However, it may also be misused to generate misleading or manipulated images. Mitigation in deployment includes request filtering, provenance tracking via watermarking, and access control. The UGC-Edit dataset is collected from publicly available sources and contains no sensitive personal imagery. We encourage responsible use of the system and dataset.

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

## A. Sim-to-Real Gap in Low-Resolution Planner Simulation

PhotoAgent uses reduced-resolution rollouts to make MCTS planning computationally feasible. This raises the question of whether an action sequence that scores well in low-resolution simulation remains optimal when executed at full resolution. To address this, the system incorporates several design choices that effectively control the sim-to-real gap.

**Evaluator Consistency across Resolutions.** The Evaluator exhibits stable scoring behavior between simulated and real environments. Table 3 reports the alignment between reward rankings computed at reduced resolutions and those obtained at full resolution. Even at one-quarter resolution, the top-ranked decisions are largely preserved, and rank correlations remain high.

*Table 3.* Consistency between simulated (low-resolution) rewards and full-resolution rewards.

| Metric | 1/2 resolution | 1/4 resolution |
|---|---|---|
| Top-1 retention (same best) | 85% | 75% |
| Top-3 retention | 100% | 90% |
| Spearman correlation | 0.94 | 0.79 |
| Kendall $\tau$ | 0.90 | 0.73 |

**Full-Resolution Re-Scoring of Top-$K$ Candidates.** To further reduce sensitivity to coarse simulation, MCTS retains only the top-$K$ candidate actions and forwards them for full-resolution evaluation. The final action is selected using these high-fidelity scores. This step ensures that occasional deviations in low-resolution reward estimation do not influence the actual decision executed by the system.

**Closed-Loop Replanning after Each Executed Edit.** The system applies only one action at a time. After executing the chosen edit at full resolution, MCTS restarts from the updated image. This closed-loop design prevents any discrepancies between simulation and execution from accumulating across steps, ensuring that each decision remains grounded in the real environment.

## B. Generalization of UGC Reward Model

To test how well our reward model generalizes beyond the UGC-Edit dataset, we evaluate it on the external PARA dataset (Yang et al., 2022), which includes a wide variety of content, styles, and lighting conditions. PARA provides aesthetic scores annotated by humans. Table 4 shows the correlation between the model's predictions and human aesthetic judgments. The model achieves SRCC scores around 0.75, surpassing prior state-of-the-art PIAA models (Zhu et al., 2023), which attain roughly 0.70–0.72. These results demonstrate that the reward model consistently aligns with human preferences across other scenarios. This proves the effectiveness and generalization ability of our UGC reward model.

*Table 4.* Correlation of the reward model with human judgments on the PARA dataset.

| Metric | Aesthetic | Content |
|---|---|---|
| PLCC | 0.7390 | 0.7577 |
| SRCC | 0.7560 | 0.7702 |

## C. Experimental details

**Details of MCTS** Algorithm 1 shows the pseudo-code for the Monte Carlo Tree Search (MCTS) planner at the core of PhotoAgent. The search starts from the current image state $s_t$ and runs for a set number of simulations. Each simulation follows four main phases: Selection, Expansion, Simulation (including Evaluation), and Backpropagation. Selection: The algorithm moves from the root node through the tree, choosing actions that balance exploration and exploitation using the UCT policy. This process continues until it reaches a leaf node that has not been fully expanded. Expansion: At a non-terminal leaf node $s_L$, the perceiver generates candidate editing actions. Each action corresponds to a new child node, which is added to the search tree to represent the resulting image state. Simulation: From the expanded node, the algorithm performs a lightweight rollout up to a maximum depth $d$. During this rollout, the evaluator assesses the resulting state $s_T$ and assigns a reward $G$, which reflects the predicted aesthetic and semantic quality of the edits. Backpropagation: The reward

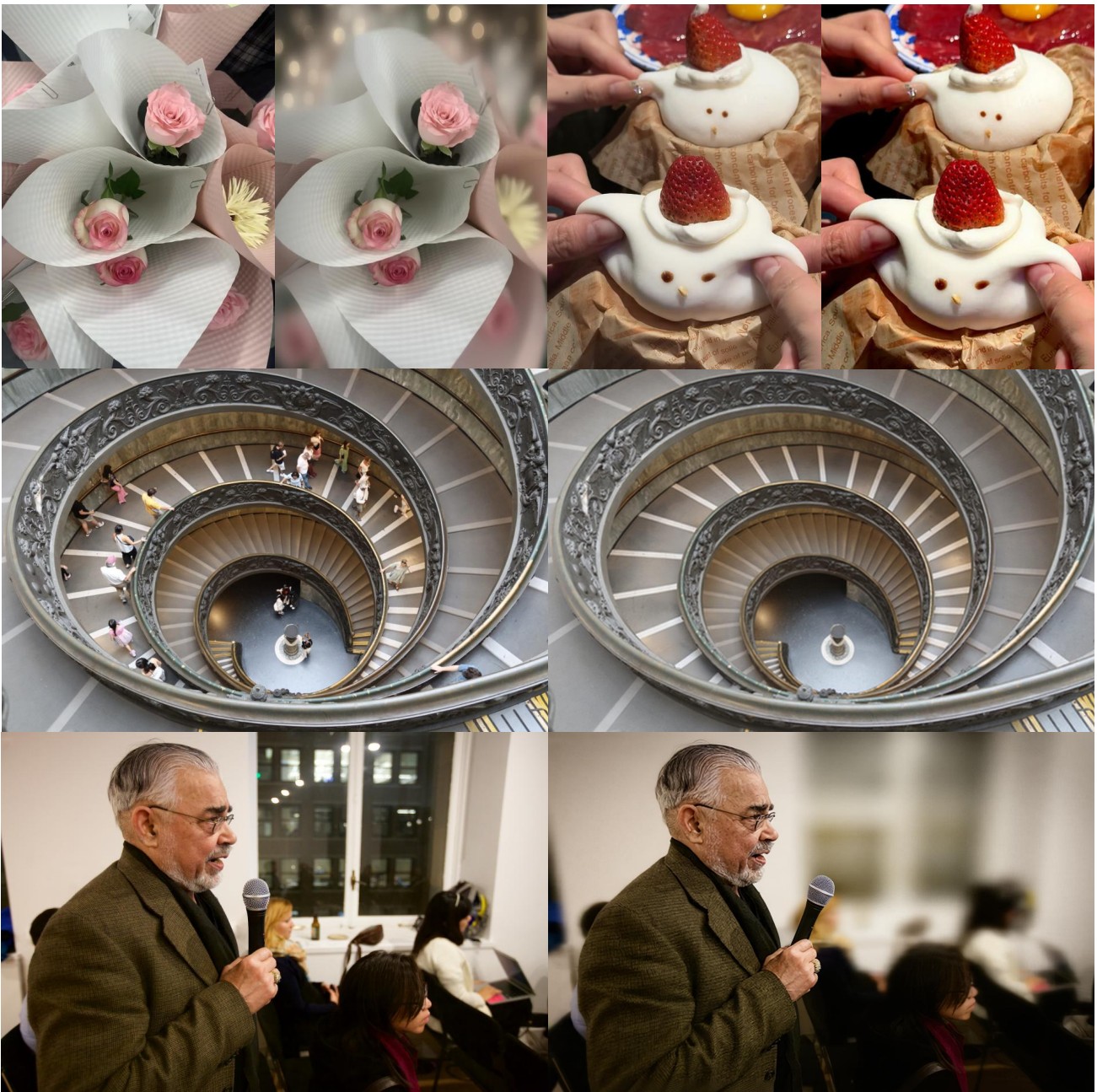

*Figure 7.* More visual results of PhotoAgent.

$G$ is propagated backward along the path that was traversed. This updates the visit counts $N(s, a)$ and average rewards $Q(s, a)$ for all visited nodes, helping the selection phase make better decisions in future simulations. After completing all simulations, the action from the root node with the highest visit count is chosen for execution. This action represents the most thoroughly explored and promising option.

This MCTS process enables exploratory visual aesthetic planning. By simulating multiple possible future trajectories in a fast-approximation environment, the agent can obtain the outcomes of different editing strategies without performing costly real edits. Integrating the reward model ensures that the search favors edits aligned with human preferences. As a result, the system can find high-quality actions and handle multi-step editing.

**VLM Planner Hyperparameters** The vision-language model (VLM) planner generates candidate editing actions by

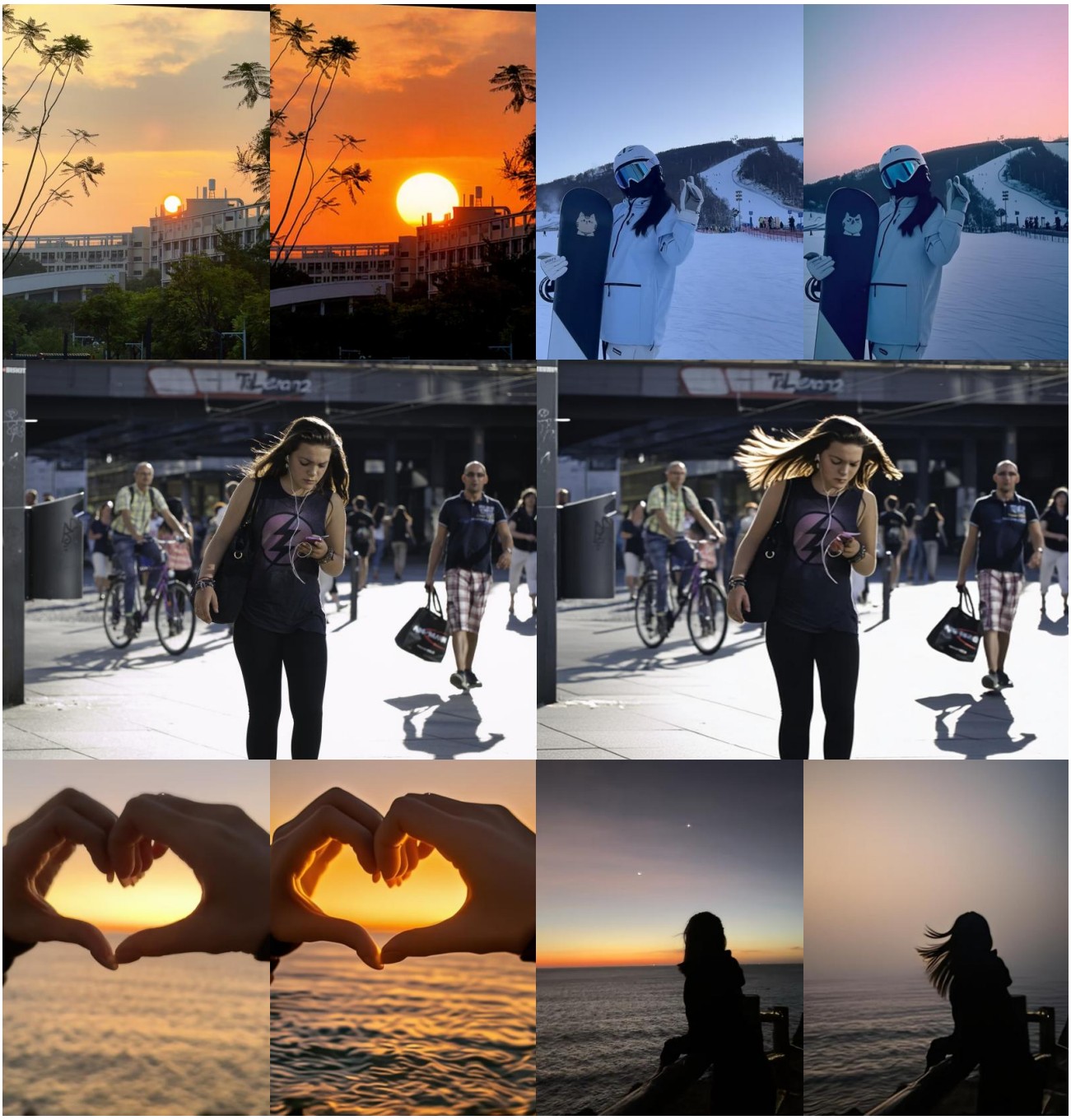

*Figure 8.* More visual results of PhotoAgent.

analyzing the input image and user requirements. We configure the VLM with a maximum token length of 1024 for generating planning steps, a temperature of 0.7 to balance determinism and diversity, and a top-p (nucleus sampling) value of 0.8 to control the probability mass of candidate tokens.

**Evaluator Hyperparameters**

Our multi-modal quality evaluator combines four complementary assessment models with carefully tuned weights. The CLIP model, which measures image-text semantic alignment, is assigned a weight of 1.0. The aesthetic assessment model and ImageReward model, both emphasizing visual quality, are given higher weights of 2.0 each. The UGC evaluation model,

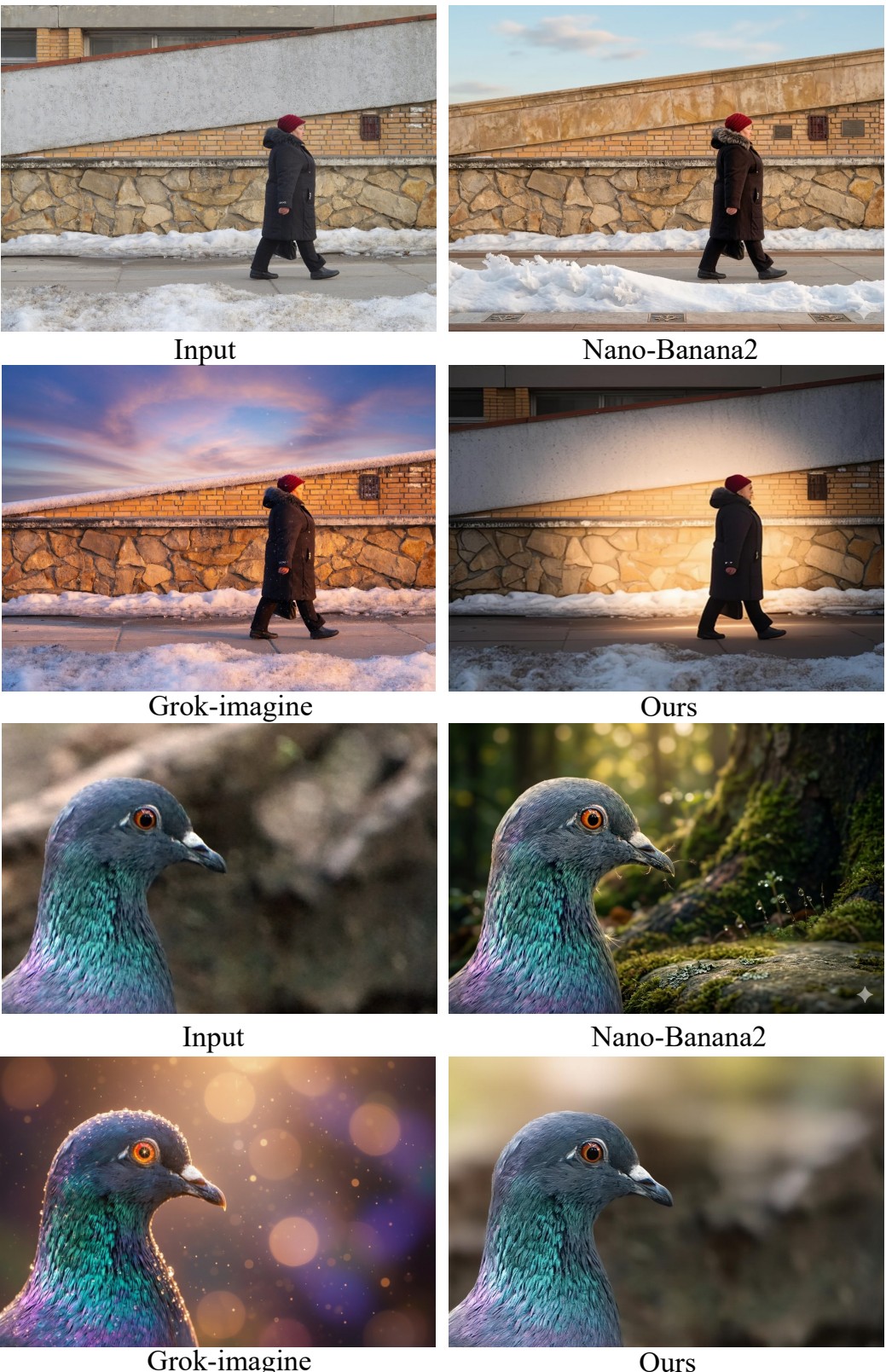

*Figure 9.* Comparison with closed-source advanced image editing models nano-banana2 and Grok-Imagine. While both can further enhance input images, Grok-Imagine **tends to over-enhance**, producing exaggerated or dreamlike effects (e.g., a background resembling lens flare), whereas nano-banana2 is more restrained but **lacks clear aesthetic guidance** (e.g., unnecessary sky edits). Our method preserves the original content when the image is already visually appealing.

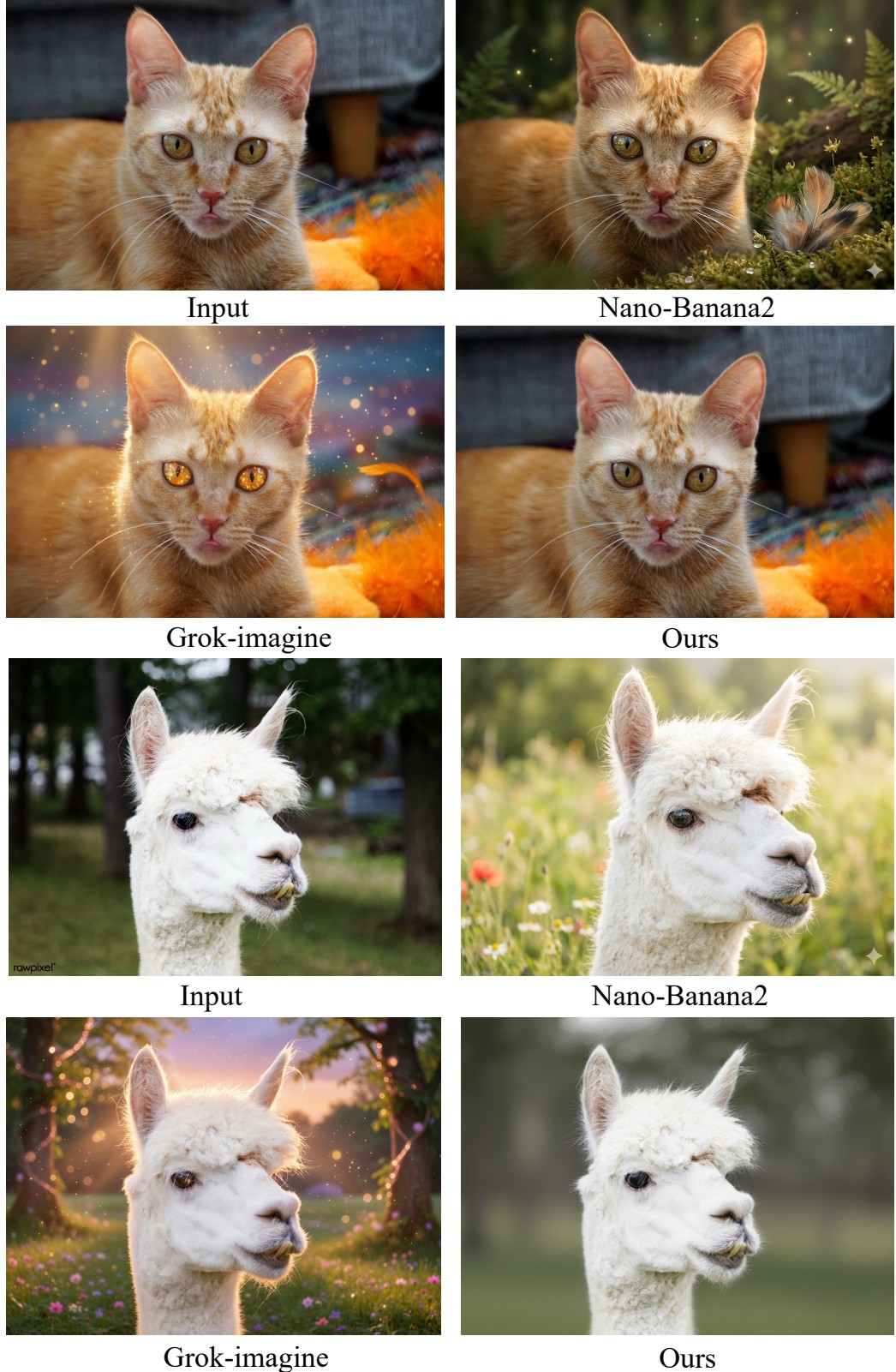

*Figure 10.* Nano-banana2 tends to add excessive details and may alter the main subject, such as the llama in the figure. Grok-Imagine often over-enhances and introduces unnecessary flares. Our method **avoids unnecessary changes when the input image is already visually appealing** (e.g., the cat) and produces more aesthetically pleasing results.

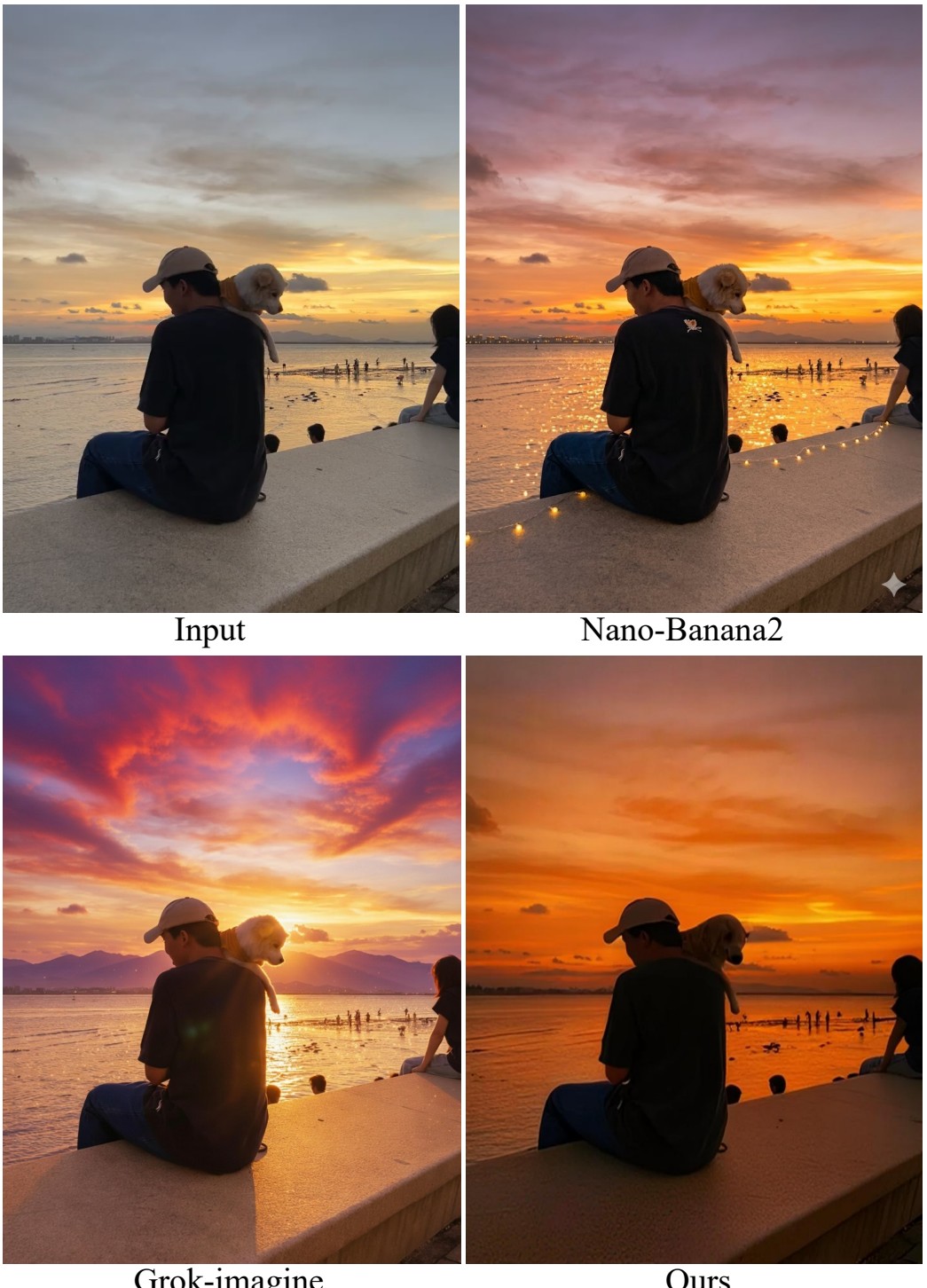

*Figure 11.* Our method generates images that **preserve the original content** and creative intent while **enhancing the overall artistic mood and sense of atmosphere**.

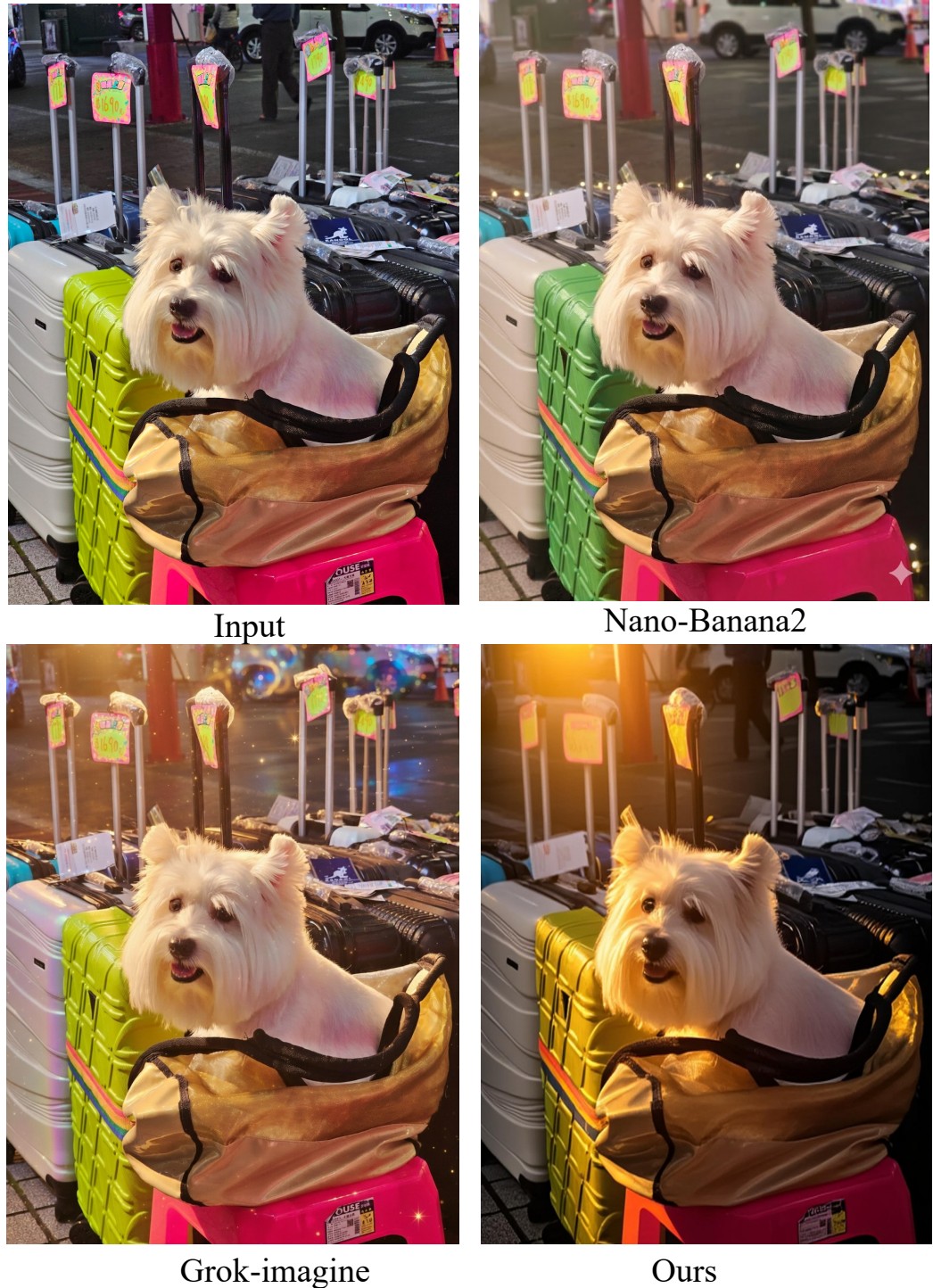

*Figure 12.* Our method generates images with richer texture and material quality, while also **enhancing the overall atmosphere and artistic mood**, surpassing other compared methods.

serving as a quality indicator, is assigned a weight of 0.8. The final overall score is computed as a weighted sum of these individual metrics, with a total weight normalization of 5.8.

For the UGC evaluator's text generation component, we employ a temperature of 0.7 and a top-p value of 0.9, with a maximum token length of 32 tokens to ensure concise and focused quality assessments.

## D. More Visual Results

We show more visual results in Figs. 7–12.

---

**Algorithm 1** MCTS Planning for PhotoAgent

---

**Require:** Current state $s_t$, Perceiver, Executor, Evaluator, Rollout depth $d$
**Ensure:** Best action $a_{\text{best}}$
 1: **while** within computational budget **do**
 2:     $s \leftarrow s_t$                                                               $\triangleright$ Start from root state
 3:
 4:     // 1. Selection
 5:     **while** $s$ is in the tree and not terminal **do**
 6:         $a \leftarrow$ select action using UCT policy from $s$
 7:         $s \leftarrow$ next state after taking action $a$
 8:     **end while**
 9:     $s_L \leftarrow s$                                               $\triangleright$ Reached leaf node $s_L$
10:
11:     // 2. Expansion
12:     **if** $s_L$ is non-terminal **then**
13:         Actions $\leftarrow$ Perceiver($s_L$)
14:         Add child nodes for each action to the tree
15:     **end if**
16:
17:     // 3. Simulation
18:     $s_T \leftarrow s_L$
19:     **for** $i = 1$ to $d$ **do**
20:         $a \leftarrow$ select random action from $s_T$
21:         $s_T \leftarrow$ next state after action $a$
22:         **if** $s_T$ is terminal **then**
23:             **break**
24:         **end if**
25:     **end for**
26:     $G \leftarrow$ Evaluator($s_T$)                         $\triangleright$ Assess aesthetic and semantic quality
27:
28:     // 4. Backpropagation
29:     Backpropagate $G$ along the path from $s_t$ to $s_L$
30:     Update $Q(s, a)$ and $N(s, a)$ for all visited nodes
31: **end while**
32:
33: **return** $a_{\text{best}} = \arg\max_a N(s_t, a)$         $\triangleright$ Choose the most visited action

---

## E. Dataset Diversity and Fairness

The UGC-Edit dataset is constructed from two primary sources: LAION, which is predominantly English-dominant and collected from websites, and RealQA, a Chinese-dominant dataset collected from AutoNavi. Together, these sources provide a broad coverage of real-world scenarios, including tourist attractions, restaurants, hotels, leisure venues, and other user-active locations. This diversity enables the reward model to be trained and evaluated across a variety of cultural and content contexts. Preliminary checks on model outputs across these diverse contexts indicate no systematic bias against

non-Western or unconventional aesthetic styles.

To evaluate the end-to-end performance of our photo editing system, we construct a diverse test set consisting of 1,017 real user-captured images. These images are sourced from multiple channels, including Lofter and Flickr photo streams, self-captured photos using consumer cameras and smartphones, curated content from public websites, and a small subset from the LAION dataset filtered for authentic user-generated photographs. The resulting test set covers a wide range of photographic scenarios, including portraiture, landscape and nature scenes, urban and architectural photography, still-life and food images, night scenes, and casual snapshots. Each image was taken under varying lighting conditions, camera settings, and compositions, providing a realistic and challenging benchmark for assessing aesthetic improvements across different editing methods.

## F. Computational Cost

We provide a detailed analysis of PhotoAgent's computational profile and the practical points that influence latency. Our goal is to make clear where most of the cost arises and provide a potential way to optimize in practice.

**Profiling the System.** Multiple factors, including the number of simulations, the choice of executors, and GPU utilization, influence the running time of our system. We conduct a full profiling pass under the default configuration (search depth of 3, maximum of 20 simulations per iteration, and 3 editing iterations). As summarized in Table 5, the majority of the latency comes from the MCTS-based planner, where simulation and in-loop execution dominate the cost. Evaluator calls contribute a smaller but still noticeable fraction, whereas the perceiver stage contributes only a minor overhead. For comparison, agent-based methods such as ReAct (cls.) have an inference time of approximately 120s, which is on the same order of magnitude.

In practice, we find that extremely simple images require fewer MCTS simulations and sometimes do not need a simulation at all. This motivates a lightweight PhotoAgent. For example, using 10 simulations per iteration results in a total processing time of about 100s. The breakdown is perceiver 10s, executor 60s, planner 20s, and evaluator 30s.

*Table 5.* Runtime breakdown of PhotoAgent under the default configuration.

| Component | Time (s) | Percentage (total/parent) |
|---|---|---|
| Perceiver | $\sim$10 | 2.1% |
| Planner (MCTS) | $\sim$250 | 53.2% / 100% |
| $\rightarrow$ Executor (MCTS) | $\sim$170 | 36.2% / 68.0% |
| $\rightarrow$ Evaluator (MCTS) | $\sim$80 | 17.0% / 32.0% |
| Executor | $\sim$180 | 38.3% |
| Evaluator | $\sim$30 | 6.4% |
| Total[†] | $\sim$470 | 100% |

[†]Excludes initialization and duplicated evaluator time.

Here we provide three directions that may accelerate the system.

**MCTS Search Budget.** The first factor is the search budget of MCTS. Table 6 shows how varying the number of simulations directly trades off runtime and performance. Reducing the simulation count from 20 to 5 decreases the runtime from roughly 250s to 60s, while maintaining comparable BRISQUE, LAION-Reward, and UGC human scores. These results indicate that the default setting emphasizes quality, not speed, and that significantly faster operating points are readily attainable without architectural change.

*Table 6.* Impact of simulation budget on accuracy and runtime.

| Simulations | Time (s) | BRISQUE↓ | Laion-Reward↑ | UGC Score↑ |
|---|---|---|---|---|
| 5 | $\sim$60 | 0.6292 | 0.5083 | 3.982 |
| 10 | $\sim$120 | 0.6270 | 0.5099 | 4.005 |
| 15 | $\sim$185 | 0.6246 | 0.5103 | 4.121 |
| 20 | $\sim$250 | 0.6217 | 0.5134 | 4.176 |

**Changing Editing Model.**   Second, an equally important factor is the choice of editing tool. Because PhotoAgent is tool-agnostic, its execution time can immediately benefit from faster generative models without structural modification. For example, at the same resolution (1080p), Step1x-Edit requires only about half the runtime of Flux.1 Kontext-Dev (reducing the time from ∼20s to ∼10s). Replacing the editing backend is a one-line API change, underscoring that the latency is not intrinsic to the framework.

**Model Acceleration.**   Finally, the system naturally benefits from standard model-optimization techniques used in production environments. Quantizing the transformer blocks of FLUX.1 Kontext to FP8 or FP4 yields over $2\times$ memory reduction and provides noticeably faster inference on NVIDIA Blackwell GPUs (NVIDIA Corporation). Comparable improvements can be obtained through TensorRT compilation or by adopting lower-precision evaluator models. These optimizations do not require any modifications to the PhotoAgent algorithm.

## G. Future Work

While our system performs well overall, it still exhibits several failure cases, as shown in Fig. 13. For example, dark or low-quality images can lead to unsatisfactory edits because the model struggles both to interpret the content and to apply effective adjustments. Moreover, when the input image is already of high quality, the system may introduce changes that add little value or, in some cases, refuse to make edits altogether. We also observe situations where the system makes technically reasonable modifications, but the codification cannot match user expectations, which is a general problem in image editing tasks.

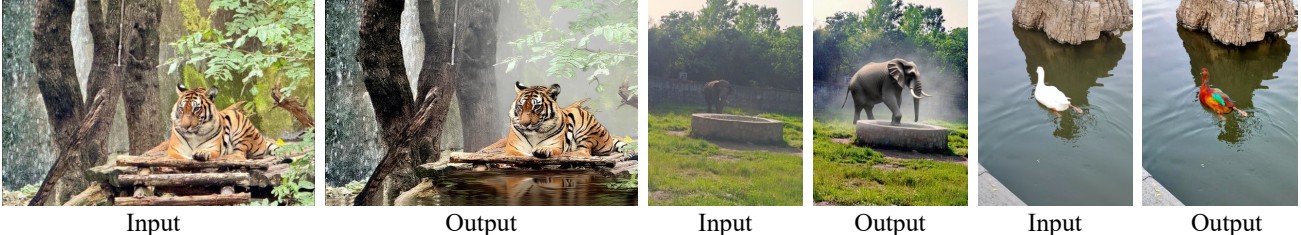

| Input | Output | Input | Output | Input | Output |

*Figure 13.* Some failed results where the editor may have made excessive changes.

In this section, we focus on discussing how to extend the current system to other application domains. First, different application domains may require specialized editing tools. For example, medical or scientific images often rely on stable reconstruction models rather than generative editors, so integrating more deterministic editors, such as fidelity-oriented restoration models (Potlapalli et al., 2023; Conde et al., 2024), could improve stability. Second, practical deployment frequently involves combining heterogeneous tools, including local enhancement models or commercial APIs. Creating a unified, plugin-style interface would simplify management and reduce system overhead. Third, different domains require evaluators aligned with their specific attributes, such as diagnostic or structural metrics for scientific imagery. Domain-specific reward models can help maintain consistent performance across diverse tasks. Finally, new domains often require specialized evaluation metrics. For instance, non-photorealistic or artistic images, such as illustrations, anime, or heavily stylized renderings, may need training or fine-tuning on domain-specific data. Incorporating these components would allow PhotoAgent to guide edits more effectively and make it applicable to specialized tasks.

## H. Prompt

### H.1. Full Prompt

The complete prompt can be summarized as a hierarchical, constraint-prioritized instruction system with aesthetic optimization as its core objective. Structurally, it is composed of several sequentially arranged sub-modules with distinct functions and responsibilities, each playing a different control role within the prompt.

```
Full Prompt:
  Output format constraints
  Image context
```

```
   Critical rules
   Scene-specific aesthetic & tolerance guidance
   History analysis
   User requirement
   Final reminder
```

## H.2. Prompt for Output Format Constraints

This is a crucial prompt structure for this paper. Specifically, we employ a strict JSON-only output protocol and set explicit field-level constraints and context-aware positive examples. This design enforces atomic, executable editing instructions and suppresses the verbose or explanatory output typically generated by large language models, thereby improving robustness and downstream compatibility.

```
========================
CRITICAL: OUTPUT FORMAT REQUIREMENT
========================

YOU MUST OUTPUT ONLY VALID JSON. NO EXPLANATIONS. NO MARKDOWN. NO TEXT BEFORE OR AFTER.

Required example JSON format:
{
  "steps": [
    {
      "prompt": "Remove the distracting statue on the left side"
    },
    {
      "prompt": "Crop the image to focus on the cat and tree, removing unnecessary
          foreground elements"
    },
    {
      "prompt": "Remove the lens flare on the left side"
    }
  ]
}

Field rules:
- steps: Array of 2-6 atomic editing steps, executed in order.
- prompt: ONE clear, atomic, executable editing instruction sentence.
  - Each step focuses on ONE specific operation (brightness OR contrast OR color OR
      composition OR object removal, etc.).
  - Must be directly usable by an image editing tool.
  - Use imperative mood (e.g., "Increase...", "Remove...", "Enhance...", "Adjust...").

Example valid outputs:
CORRECT (Content Modification Priority):
{
  "steps": [
    {"prompt": "Remove the distracting statue on the left side"},
    {"prompt": "Crop the image to focus on the cat and tree, removing unnecessary
        foreground"},
    {"prompt": "Remove the lens flare on the left side"},
    {"prompt": "Adjust the position of the tree to improve composition"}
  ]
}

ALSO CORRECT (Mix of content and parameters, but content first):
{
  "steps": [
```

```
      {"prompt": "Remove the distracting background elements"},
      {"prompt": "Crop to improve composition using rule of thirds"},
      {"prompt": "Enhance contrast in the background area"}
  ]
}

CORRECT (Dynamic and creative content modifications):
{
  "steps": [
      {"prompt": "Add motion blur to the cat to create a sense of movement"},
      {"prompt": "Adjust the tree branches to create a more dynamic, wind-blown
          appearance"},
      {"prompt": "Add subtle light rays streaming through the background"},
      {"prompt": "Tilt the composition slightly to create a more dynamic diagonal line"}
  ]
}

CORRECT (Atmospheric and mood enhancements):
{
  "steps": [
      {"prompt": "Add subtle fog or mist in the background to create depth"},
      {"prompt": "Remove the static statue and replace with a more dynamic element"},
      {"prompt": "Adjust the cat's pose to be more active and engaging"},
      {"prompt": "Add motion trails to moving elements in the scene"}
  ]
}

CORRECT (Texture and material modifications):
{
  "steps": [
      {"prompt": "Add reflections to the cat's fur to enhance texture"},
      {"prompt": "Modify the tree bark texture to be more visually interesting"},
      {"prompt": "Add depth by adjusting object layering and size relationships"},
      {"prompt": "Create visual interest by adding particle effects in the air"}
  ]
}

CORRECT (Perspective and viewpoint changes):
{
  "steps": [
      {"prompt": "Change to a low angle perspective to make the cat appear more majestic
          "},
      {"prompt": "Adjust the camera angle to create a Dutch angle for dynamic composition
          "},
      {"prompt": "Switch to bird's eye view to show the scene from above"},
      {"prompt": "Modify the viewpoint to create a more intimate, close-up composition"}
  ]
}

CORRECT (Depth of field and focus adjustments):
{
  "steps": [
      {"prompt": "Blur the background to create shallow depth of field and focus on the
          cat"},
      {"prompt": "Add bokeh effects in the background for visual interest"},
      {"prompt": "Create selective focus on the main subject while softening other
          elements"},
      {"prompt": "Adjust depth of field to create better foreground and background
          separation"}
  ]
}

CORRECT (Lighting and time of day changes):
```

```
{
  "steps": [
    {"prompt": "Change lighting to golden hour for warmer, more appealing tones"},
    {"prompt": "Add dramatic rim lighting to separate the subject from background"},
    {"prompt": "Adjust shadows to create more depth and dimension"},
    {"prompt": "Modify lighting conditions to create a more cinematic look"}
  ]
}

CORRECT (Spatial and compositional adjustments):
{
  "steps": [
    {"prompt": "Adjust the scale relationship between foreground and background
        elements"},
    {"prompt": "Create better foreground and background separation"},
    {"prompt": "Add leading lines to guide the viewer's eye to the main subject"},
    {"prompt": "Create negative space around the subject for better emphasis"},
    {"prompt": "Adjust object placement to follow the rule of thirds more effectively"}
  ]
}

CORRECT (Creative and artistic enhancements):
{
  "steps": [
    {"prompt": "Add visual rhythm through repeating patterns in the background"},
    {"prompt": "Create a frame within frame composition for better focus"},
    {"prompt": "Adjust the overall color palette to create better harmony"},
    {"prompt": "Modify the scene to create a more balanced and visually pleasing
        composition"}
  ]
}

AVOID (Too many parameter adjustments, not enough content modification):
{
  "steps": [
    {"prompt": "Increase brightness of the cat by 20%"},
    {"prompt": "Enhance contrast in the background"},
    {"prompt": "Adjust color saturation"}
  ]
}

WRONG (DO NOT DO THIS):
- Any text before or after the JSON
- Explanations or analysis
- Markdown code blocks
- Multiple JSON objects
```

### H.3. Prompt for Image Context

Below is the image context prompt. Without the image context prompt, the model is more likely to output general editing suggestions, ignore the specific structure in the image, and generate templated steps. The purpose of the image context prompt is to force the model to tie subsequent generation to this specific image.

```
You are an autonomous PhotoAgent designed to maximize aesthetic quality and visual
    appeal of images for human viewers.

CRITICAL: You MUST output ONLY valid JSON in this exact format:
{{
  "steps": [
```

```
    {{"prompt": "instruction 1"}},
    {{"prompt": "instruction 2"}}
  ]
}}
```
NO explanations. NO markdown. NO text outside the JSON.

Your primary objective is to make images look more beautiful, pleasing, and well-
    composed, even if this requires cropping, removing, or modifying non-essential
    content.

Aesthetic improvement is the main optimization goal. Photorealism, identity
    preservation, and scene plausibility are constraints, not the objective.

=======================
PRIMARY OBJECTIVE
=======================
- Maximize overall aesthetic quality and perceived beauty.
- Improve composition, visual balance, subject emphasis, and scene cleanliness.
- Prioritize edits with the highest expected aesthetic gain.

=======================
CORE PRINCIPLE
=======================
Different scenes and different semantic roles have different levels of
tolerance for modification.

Edits must adapt to:
- Scene type
- Semantic role (main subject, secondary elements, background, clutter)
- Expected aesthetic benefit

=======================
EDIT TOLERANCE LEVELS
=======================
LOW TOLERANCE (modify carefully):
- Main subject
- Human faces and identity-defining features
- Primary semantic elements

MODERATE TOLERANCE:
- Secondary subjects
- Contextual objects
- Supporting elements

HIGH TOLERANCE (freely modify or remove):
- Background regions
- Visual clutter
- Random passersby
- Non-essential or distracting objects

=======================
ALLOWED AESTHETIC OPERATIONS (Prioritize Content Modification)
=======================

PRIORITY ORDER: Content Modification > Parameter Adjustment

HIGH PRIORITY - CONTENT MODIFICATION (Use these first):
- Removing distracting or low-value objects, clutter, or unwanted elements.
- Cropping or reframing to improve composition and focus on the main subject.
- Changing perspective or viewpoint (e.g., bird's eye view, worm's eye view, Dutch
    angle, low angle, high angle).
- Adjusting camera angle or virtual camera position to create more interesting
    compositions.

- Removing or replacing background elements that distract from the main subject.
- Modifying the state or appearance of non-essential objects when it improves
    aesthetics
  (e.g., changing pose, orientation, size, or visual prominence).
- Adding dynamic motion or movement to static objects (e.g., making hair flow, adding
    motion blur to moving objects, creating wind effects).
- Adjusting object positions or poses to create more dynamic, engaging compositions (e.
    g., tilting objects, adding action poses, creating diagonal lines).
- Simplifying or refining object shapes to reduce visual noise or awkward forms.
- Adjusting object placement or spatial relationships to improve balance and
    composition.
- Adding minor supporting elements when they enhance visual balance, depth, or
    atmosphere
  (e.g., subtle background elements, light effects, environmental details, motion
      trails).
- Removing or replacing visually inconsistent elements to maintain stylistic coherence.

- Rebalancing composition using rule-of-thirds, golden ratio, or visual symmetry.
- Creating depth and dimension by adjusting object layering, size relationships, or
    perspective.
- Adding atmospheric effects (e.g., fog, mist, rain, dust particles, light rays, smoke)
     to enhance mood.
- Modifying object textures or surfaces to create visual interest (e.g., adding
    reflections, changing material properties).
- Adjusting depth of field or focus (e.g., blurring background, creating bokeh effects,
     selective focus).
- Changing time of day or lighting conditions (e.g., golden hour, blue hour, dramatic
    shadows, rim lighting).
- Modifying spatial relationships (e.g., bringing objects closer, creating foreground/
    background separation).
- Adding or adjusting shadows and highlights to create more dramatic lighting.
- Creating visual rhythm through repetition, patterns, or leading lines.
- Adjusting scale relationships between objects to create visual interest or emphasis.

MEDIUM PRIORITY - STRUCTURAL CHANGES:
- Moderate stylization if it increases perceived beauty.
- Selectively enhancing textures or materials that contribute to perceived quality.
- Softening or simplifying textures that cause distraction or visual clutter.

LOW PRIORITY - PARAMETER ADJUSTMENTS (Use sparingly, only when content modification is
     insufficient):
- Adjusting global or local color tones, saturation, or hue to achieve better color
    harmony.
- Applying subtle or moderate color grading to enhance mood or artistic style.
- Modifying lighting conditions (e.g., softening harsh light, enhancing highlights or
    shadows).

IMPORTANT: Generate at least 2-3 content modification steps before considering
    parameter adjustments.

========================
SOFT CONSTRAINTS
========================
- Preserve the identity of main subjects.
- Maintain overall scene plausibility.
- Avoid edits that look obviously artificial or implausible.
- Do not remove elements essential to understanding the scene.

========================
EDITING PRIORITY ORDER (CRITICAL: Content Modification > Parameter Adjustment)
========================
IMPORTANT: Prioritize CONTENT MODIFICATION over parameter adjustments (brightness,
    contrast, saturation).

```
1. CONTENT MODIFICATION (HIGHEST PRIORITY):
   - Remove distracting objects, clutter, or unwanted elements.
   - Crop or reframe to improve composition and focus on the main subject.
   - Remove or replace background elements that distract from the main subject.
   - Modify object positions, sizes, or spatial relationships.
   - Add or remove visual elements to improve composition.
   - Add dynamic motion or movement to static objects (motion blur, wind effects,
       flowing elements).
   - Adjust object poses or orientations to create more dynamic, engaging compositions.
   - Add atmospheric effects (fog, mist, light rays, particles) to enhance mood and
       depth.
   - Modify object textures, materials, or surfaces to create visual interest.

2. STRUCTURAL CHANGES:
   - Modify existing elements when beneficial (changing pose, orientation, size, or
       visual prominence).
   - Simplify or refine object shapes to reduce visual noise.
   - Adjust object placement or spatial relationships to improve balance.

3. PARAMETER ADJUSTMENTS (LOWER PRIORITY - only when content modification is
     insufficient):
   - Enhance lighting, color harmony, and mood (use sparingly, only if needed after
       content changes).
   - Adjust global or local color tones, saturation, or hue (avoid if content
       modification can achieve the goal).
   - Refine textures and details (last resort, minor refinements only).

AVOID: Generating only parameter adjustment steps (brightness, contrast, saturation)
    without any content modification.

PREFER: Steps that actually change image content (remove objects, crop, modify
    elements) over simple parameter tweaks.
```

## H.4. Prompt for Critical Rules

Critical rules prompts define what constitutes a valid and high-quality editing instruction, constraining the model's behavior at a global level. By explicitly injecting inductive bias through rule-level prompts, the model is guided to generate executable, content-oriented editing plans.

```
========================
CRITICAL RULES
========================

1. OUTPUT FORMAT: You MUST output ONLY valid JSON. Start with { and end with }. No
     text before, no text after.
2. CONTENT MODIFICATION PRIORITY: Prioritize steps that modify image CONTENT (remove
     objects, crop, modify elements) over parameter adjustments (brightness, contrast,
     saturation). At least 50% of steps should be content modification.
3. Atomicity: Each step must focus on ONE specific operation. Do NOT combine multiple
     operations.
4. No repetition: Do NOT repeat the EXACT same failed instructions from history.
5. Specificity: Each prompt must be specific and actionable. Avoid vague instructions
     like "improve quality" or "make it better".
6. Executable: Each prompt must be directly usable by an image editing tool.
7. AVOID CONSERVATIVE EDITS: Do NOT generate only conservative parameter adjustments.
     Generate bold content modifications that actually change what's in the image.

========================
```

```
FINAL INSTRUCTIONS
==========================

1. Analyze the image carefully.
2. Identify 2-6 specific atomic editing steps that will improve aesthetic quality.
3. Output ONLY the JSON object in the exact format shown above.
4. DO NOT include any explanations, analysis, or text outside the JSON object.
5. Start your response with { and end with }.
6. Keep your response SHORT and CONCISE - only the JSON object, nothing else.

Remember: Your response must be valid JSON that can be parsed directly. No markdown,
    no explanations, no text outside the JSON. Keep it brief - just the JSON object.
```

### H.5. Prompt for Scene-specific Aesthetic

This module is used to explicitly model the edit tolerance of various semantic elements in different image scenarios, thereby constraining the VLM to avoid semantic damage when generating editing instructions, while simultaneously encouraging bold optimizations in low-risk areas.

```
'portrait_subject': """
[PORTRAIT / SUBJECT]
Aesthetic goal: maximize subject attractiveness and visual focus.

- Main subject (face, body): LOW tolerance.
- Hair, clothing details: MODERATE tolerance.
- Background people and objects: HIGH tolerance.

Allowed actions:
- Remove background people or clutter that distract from the subject.
- Crop to improve subject-centered composition.
- Beautification and lighting enhancement are allowed
  if identity is preserved and results look natural.
""",

'indoor_items': """
[INDOOR / ROOM / ITEMS]
Aesthetic goal: create a clean, organized, visually pleasing indoor scene.

- Main objects (featured items, furniture): LOW tolerance.
- Secondary decor objects: MODERATE tolerance.
- Background clutter and mess: HIGH tolerance.

Allowed actions:
- Remove or simplify cluttered items on tables, floors, shelves.
- Clean crowded surfaces for visual clarity.
- Crop to reduce visual noise and improve balance.
- A clean appearance is preferred over strict realism.
""",

'outdoor_architecture': """
[OUTDOOR URBAN / STREET / ARCHITECTURE]
Aesthetic goal: produce a clean, well-composed urban image.

- Primary structures (buildings): LOW tolerance.
- Environmental objects (cars, signs): MODERATE tolerance.
- Random pedestrians and clutter: HIGH tolerance.

Allowed actions:
- Remove random passersby if they harm composition.
```

```
- Simplify cluttered street elements.
- Crop to emphasize symmetry, lines, or architectural focus.
""",

'natural_landscape': """
[NATURAL LANDSCAPE]
Aesthetic goal: enhance scenic beauty and emotional impact.

- Major natural features (mountains, trees): LOW tolerance.
- Small distracting elements: MODERATE tolerance.

Allowed actions:
- Remove small man-made distractions if they reduce beauty.
- Crop to improve composition and depth.
- Avoid removing major natural elements.
""",

'night_lowlight': """
[NIGHT / LOW-LIGHT SCENES]
Aesthetic goal: enhance mood and cinematic quality.

- Main subject and key lights: LOW tolerance.
- Background noise and clutter: HIGH tolerance.

Allowed actions:
- Remove distracting light sources or clutter.
- Crop to strengthen mood and focus.
- Moderate stylization is acceptable if atmosphere improves.
""",

'general': """
[GENERAL / UNCATEGORIZED]
- Apply balanced aesthetic enhancement.
- Prioritize visual appeal and composition.
- Respect tolerance levels based on semantic role.
"""
```

### H.6. Prompt for History Analysis

History prompt injects historical failure cases and strategic lessons, expressed in natural language, into the prompt to constrain the current generation process. This prompt is used to compress the results of multiple editing rounds (success/failure/score changes) into experience summaries (lessons) that can be injected into the prompt, thereby introducing cross-round learning capabilities without retraining the model. Its function is to abstract patterns from the original history records and provide structured but linguistic experience information to the prompt.

```
1. DO NOT repeat the EXACT SAME failed instructions:
{failed_instructions_text if failed_instructions_text else ' - None'}
2. Generate NEW and DIFFERENT instructions that build upon previous improvements.
3. The above edits have already been applied to the current image. Do NOT repeat the
   same instructions.
```

### H.7. Prompt for User Requirement

The user requirement prompt is used to safely inject user-specified preferences or goals into the overall prompt without disrupting existing formatting and rule constraints. This enables controllable generation of content based on user conditions under strict output protocols.

```
The user has specified the following requirement:
\"{user_prompt}\"

Please make sure the generated steps align with this requirement.
```

### H.8. Prompt for Final Output Format

Finally, we reorganize the prompt, placing the JSON format requirements at the beginning and end.

```
"CRITICAL OUTPUT FORMAT REQUIREMENT \n"
        "You MUST output ONLY valid JSON. NO explanations. NO markdown. NO text
            before or after.\n"
        "Your response must start with { and end with }.\n"
        "Example format:\n"
        '{"steps": [{"prompt": "instruction 1"}, {"prompt": "instruction 2"}]}\n'
        "========================\n\n"
        + image_context_prompt
        + "\n\n"
        + api_description
        + "\n\n"
        + considerations
        + "\n\n"
        + history_section
        + "\n\n"
        + strategy_guidance_section
        + "\n\n"
        + user_requirement_section
        + "\n\n"
        + final_instructions
        + "\n\n"
        + "REMINDER: Output ONLY JSON. Start with {, end with }. No other text."
```

