# OpenReview forum: "PhotoAgent: Exploratory Visual Aesthetic Planning with Large Vision Models"
_ICML.cc/2026/Conference — ICML 2026 spotlight_

### Official Review · Reviewer_LHfN · 2026-02-28

**Soundness:** 3
**Presentation:** 4
**Significance:** 3
**Originality:** 2
**Overall Recommendation:** 5
**Confidence:** 4

**Summary:**

This paper proposes PhotoAgent, a system enabling autonomous image editing by employing a loop of four processes: Perceiver, Planner, Executor, and Evaluator. The framework allows users to generate high-quality edited images without requiring expertise or manual effort in execution or evaluation. The authors construct the UGC-Edit dataset, consisting of 7,000 user-generated photos extracted from existing datasets (LAION and RealQA), to enable thorough evaluation of image editing systems. In human evaluations, PhotoAgent's outputs received more votes than those of competing systems.

**Compliance With Llm Reviewing Policy:**

Affirmed.

**Final Justification:**

The explanation of the region-specific tolerance levels was very clear. I highly appreciate that this is an important element and that it has already been realized. The authors' thorough rebuttal helped deepen my understanding of the method.

**Key Questions For Authors:**

- I would like the authors to provide a clear definition of what is meant by "image editing." While changes such as brightness adjustment or removal of unwanted objects may generally be acceptable, I feel uneasy about manipulations mentioned in the paper, such as altering the shape of hair, changing the direction of animals, or modifying wave heights. Do the authors consider that their method can be applied to any kind of image editing application?
- Is it not problematic that the user's initial request is not reflected in the Evaluator? For example, if a user requests that faces should not be changed, can the method guarantee a good editing result without evaluating such user constraints?
- How should the balance be maintained between the main loop and the lightweight loop inside the MCTS?

**Limitations:**

The discussion of potential negative societal impact in the paper is not sufficient. While the risks associated with image editing and manipulation are generally acknowledged, the paper does not offer any means to prevent potential misuse by malicious users. Therefore, there is a possibility that problems could arise.

**Strengths And Weaknesses:**

By appropriately combining suitable tools, the system achieves the desired functionalities to a certain extent. However, there are concerns regarding image manipulations that may go beyond what is acceptable for image editing. For example, in some cases involving humans, the hair becomes disheveled or the orientation changes, thus altering the original content of the photo. Depending on the application, such transformations may be unacceptable. It is necessary to have a method to suppress such edits.

Another issue is that the initial prompt provided by the user only affects the selection of editing methods and is not reflected in the Evaluator, which may pose a problem.

In the experiments, the usefulness of the proposed approach is shown in comparison with GPT-4o, but since GPT-4o is a relatively low-level editing tool, it may not be an adequate baseline. It remains unclear whether the proposed method is superior to nano banana or grok, which are actually used in the system.

The paper is easy to read, and all essential information is well summarized. The overview figure greatly aids in understanding the content of the paper. However, the leaf shadow in the background of the graph in Figure 2 seems unnecessary. The appendix is also extensive and useful, particularly in confirming the necessity and validity of MCTS.

Proposing a loop-based framework has significance for the area of image editing. While its applicability may be general and could extend to other domains, this makes it somewhat less original. Nevertheless, it is commendable that the appropriate tools were chosen to design and propose an integrated system.

---

> ### Author Rebuttal · Authors · 2026-03-31
>
> We sincerely thank the reviewer for the positive assessment of our work and insightful comments. We hope our response below can address the reviewer's concerns.
>
> > 1. ...there are concerns regarding image manipulations that may go beyond what is acceptable for image editing...have a method to suppress such edits.
>
> We agree that “manipulation” can be a somewhat ambiguous concept. In practice, we design region-specific tolerance levels for portrait scenes (Appendix H): the main subject (face and body) is assigned low tolerance, hair and clothing receive moderate tolerance, and background elements have high tolerance. This allows the system to perform necessary edits while preserving critical content. For example, in Fig. 4 (last row), the boy’s facial features remain unchanged despite other modifications.
>
> Moreover, our method allows users to explicitly specify preservation constraints, which is similar to negative prompts, indicating which attributes (e.g., the face) should remain unchanged. The framework leverages these constraints both to suppress unnecessary edits during generation and evaluation to protect attributes remain unchanged after editing.
>
> > 2. The initial prompt provided by the user only affects the selection of editing methods and is not reflected in the Evaluator...can the method guarantee a good editing result without evaluating such user constraints?
>
> We thank the reviewer for pointing this out. The initial user prompt is not used only for action generation but also guide the Evaluator through text-image alignment signals such as CLIP-based scoring. In addition, we design region-specific tolerance levels for portrait scenes (Appendix H), which is discussed in detail in Question 5.
>
> > 3. Whether the proposed method is superior to nano banana or grok.
>
> We thank the reviewer for this suggestion. Based on our experiments, our method consistently outperforms Nano Banana and Grok in terms of fidelity, naturalness, and aesthetic quality. Concretely, under ambiguous instructions (e.g., “make it more beautiful”), single-step methods such as Nano Banana or Grok often produce inconsistent or suboptimal edits, including over-enhancement, loss of important details, or unintended modifications to the main subject. In contrast, our framework iteratively explores and refines candidate edits, which improves content preservation and overall aesthetic coherence. Visual results are available at https://anonymous.4open.science/r/icml-photoagent-3FE9 (**please download repository for visualization**).
>
> > 4. Paper is easy to read but leaf shadow in Figure 2 appear unnecessary.
>
> We appreciate the suggestion and we will revise the figure accordingly to make it cleaner.
>
> > 5. The clear definition of “image editing” and its acceptable boundaries.
>
> We thank the reviewer for raising this point. In this work, we use “image editing” in a broad operational sense, including both low-level adjustments and high-level semantic edits accepted by the user. The system aims to preserve the photographer's intent while maximizing aesthetic quality. To clarify its boundary, our framework includes two layers of safeguards:
>
> **System-level constraints**: We set different tolerance levels for different regions, such as the main subject (face and body) is assigned low tolerance, hair and clothing receive moderate tolerance, and background elements have high tolerance (Appendix H).
>
> **User-specified constraints**: We allow users to explicitly indicate which content should remain unchanged through prompts. This guides the perceiver to avoid generating such actions. These constraints are also enforced by the evaluator, such as a CLIP-based module, to ensure that the final edits comply with the user’s intentions.
>
> Overall, technically, our framework can be applied to most image editing tasks by adjusting the planning, execution, and evaluation tools as needed. But in practice, the user serves as the final decision-maker, determining whether to accept the system’s output, while challenging cases (e.g., legal or forensic images) are left for future work.
>
> > 6. How should the balance be maintained between the main loop and the lightweight loop inside the MCTS?
>
> The lightweight MCTS loop is used for efficient lookahead and action ranking, whereas the main loop remains the final decision layer through full-resolution execution and evaluation. In practice, the balance is controlled by the MCTS search budget (e.g., simulation count and depth). As shown in Table 5, stronger planning budgets consistently improve quality, while smaller budgets provide a useful efficiency-quality trade-off.
>
> > 7. The discussion of potential negative societal impact...offer means to prevent potential misuse by malicious users.
>
> We thank the reviewer for raising this concern. In practice, mitigating misuse would require safeguards such as request filtering, restrictions on sensitive edits, and verification mechanisms. We will expand this discussion in the revision.

---

> > ### Author Rebuttal · Reviewer_LHfN · 2026-04-03
> >
> > I have thought from the beginning that this paper is worthy of acceptance. In particular, the explanation of the region-specific tolerance levels was very clear. I highly appreciate that this is an important element and that it has already been realized. If I were to increase the Overall score, it would have to go up to the maximum of 6, but I cannot praise it to that extent, so I will keep the previous Overall score. I will increase the Significance score by one point.
> >
> > For the following response, what I wanted to know is still not completely clear. In Figure 2, the Edited image is sent from the Evaluator to the Perceiver. Users’ prompts may sometimes imply that certain metrics should be ignored. It seems more convincing if the score itself is passed to the Perceiver and taken into account when constructing the action list.
> >
> > "The initial user prompt is not used only for action generation but also to guide the Evaluator through text–image alignment signals such as CLIP-based scoring."
> >
> > I also appreciate, as a reviewer, that the authors compared the proposed method against Nano Banana and Grok. Intuitively, the proposed method appears at least comparable, but there were some samples where I was not entirely convinced that it is consistently superior (even though I assume the authors selected good examples to showcase).

---

> > > ### Author Response · Authors · 2026-04-08
> > >
> > > We sincerely thank the reviewer for the high recognition of our work, which is very encouraging.
> > >
> > > We further clarify the details of what the Perceiver receives. **First**, the Perceiver currently support accepting historical editing information (see Appendix H), which ensures that the new action list takes previous edits into account. **Second**, if users have specific requirements for certain metrics, they can manually adjust the relative importance of different metrics or remove certain metrics entirely. This is more controllable and precise than specifying it directly in the prompt, avoiding execution deviations caused by ambiguities in prompt language understanding.
> > >
> > > We again appreciate the reviewer's feedback, which has helped us improve our method. If you have any further questions, please feel free to raise them at any time.

---

### Official Review · Reviewer_mFre · 2026-03-07

**Soundness:** 3
**Presentation:** 3
**Significance:** 2
**Originality:** 2
**Overall Recommendation:** 3
**Confidence:** 4

**Summary:**

PhotoAgent is an autonomous image editing system with multiple components: perceiver, planner, executor, evaluator. The system leverages LLMs, image editing models and other tools. The paper also introduces UGC-Edit, an aesthetic evaluation benchmark of 7000 photos. Lastly, the paper compares current methods on various editing scenarios.

**Compliance With Llm Reviewing Policy:**

Affirmed.

**Final Justification:**

The rebuttal addressed my concerns regarding the quality and execution. However I am still unsure of the long-term impact of this work and it reads like mostly an engineering integration. My score is updated to a 3.

**Key Questions For Authors:**

Why are components of the PhotoAgent system (such as Show-O, Bagel or NanoBanana) not included in the evaluation? Including them would clarify how the method compares to current state-of-the-art approaches.
Who were the annotators, how were they instructed, and how was evaluation consistency ensured?
What is the novelty of the proposed dataset?

**Limitations:**

yes

**Strengths And Weaknesses:**

Strengths:
The paper frames image editing as a long-horizon process rather than a single-step transformation, which a useful perspective for complex editing workflows. The system is evaluated against several baselines and includes a human evaluation. Empirically, the proposed system achieves stronger benchmark performance than the compared methods. The work also introduces a new dataset that may be useful for future research on multi-step image editing.

Weaknesses:

1. The paper largely reads as an engineering integration of existing tools, and it is unclear what scientific insight emerges from this integration. It is difficult to determine what new research questions this work enables.

2. The evaluation relies primarily on older baselines, with most methods dates back to 2023. At the same time, the executor appears to rely on more modern tools such as Show-O and NanoBanana, yet these systems are not included as baselines in the comparison. This makes it difficult to assess how the proposed approach performs relatively to its components. Including stronger and more recent baselines would provide a more meaningful and informative evaluation.

3. CLIP-based metrics are known to correlate poorly with human judgment. The evaluation would benefit from using VLM-as-a-judge metrics. The user study is also underdescribed: the paper does not specify who the participants are, how they were recruited, or what instructions they received.

4. The dataset construction raises concerns. The 7K dataset is derived from LAION, which is known to contain substantial noise, and the use of an “aesthetic” filter introduces subjective and culturally dependent biases. The paper does not clearly compare this dataset to existing ones or explain what unique features it contributes.

5. The reward model is both trained and evaluated on the same dataset, and external validation is not sufficiently justified. This makes it unclear whether the improvements reflect real quality gains or overfitting.

6. Efficiency is also a concern. The appendix states that the system requires more than 500 seconds per edit, raising questions about the practicality of the approach.

---

> ### Author Rebuttal · Authors · 2026-03-31
>
> > 1. The paper reads as an engineering integration of existing tools, and it is unclear what scientific insight emerges.
>
> Directly integrating existing image editing tools is not sufficient as these tools are loosely coupled and lack coordination, and off-the-shelf vision-language models are not designed for image editing. Naïve one-shot calls often lead to short-sighted decisions and irreversible edits. Instead, we found it more effective to formulate image editing as a long-horizon planning problem, treating the modules as an integrated system optimized via multi-step exploration and closed-loop feedback. To the best of our knowledge, this is the first work to formulate image editing in this way.
>
> This framing yields three insights: (1) planning matters, as exploration consistently outperforms reactive/open-loop baselines; (2) reduced-resolution simulation is an effective planning proxy; and (3) we introduce a UGC editing dataset that enables reward alignment for aesthetic preferences in user-generated content.
>
> > 2. More executors should be included as baselines.
>
> We thank the reviewer for this suggestion. We have added grok, and NanoBanana2.0 as standalone single-step baselines (https://anonymous.4open.science/r/icml-photoagent-3FE9, **please download repository for visualization**). The results show that PhotoAgent consistently outperforms baselines. Concretely, under ambiguous instructions (e.g., “make it more beautiful”), single-step methods often produce inconsistent or suboptimal edits, such as over-enhancement, loss of important details, or unintended changes to the main subject. In contrast, PhotoAgent can iteratively explore and refine candidate edits, leading to better content preservation and improved overall aesthetic coherence.
>
> > 3. CLIP-based metrics correlate poorly with human judgment and VLM-as-a-judge would be better. User study is underdescribed.
>
> In our evaluation, CLIP similarity is used for content preservation. To better reflect human preference, we also include several complementary evaluators, including a VLM-based UGC reward model. As shown in Appendix B, this model correlates better with human judgments on the external PARA benchmark than generic metrics such as ImageReward and LAION-Aesthetic.
>
> For details of user study, participants range from casual smartphone users to professional photographers. They were recruited through internal announcements and volunteered to participate. In each trial, they see the input image and the outputs of all four methods in randomized, unlabeled order. Participants are instructed to: "Select the image with the best editing result. The criteria are: while preserving the original photo content as much as possible, the edited image should be more aesthetically pleasing overall and something you would be more willing to share.'' We will clarify these details in the revision.
>
> > 4. Is there noise, subjective and culturally dependent biases of the UGC datasets. What unique features it contributes?
>
> Thank you for raising this concern. **We have discussed Dataset Diversity and Fairness in Appendix E** and we further clarify it from three aspects.
>
> **Noise/Subjectivity**. We do not use raw LAION directly. Instead, UGC-Edit is curated through a two-stage pipeline (Section 4): VLM-based filtering followed by human verification. This step mainly removes clearly non-UGC content (e.g., posters, screenshots, advertisements, and AI-generated images), which is relatively objective and less dependent on subjective judgment.
>
> **Cultural Bias**. UGC-Edit is constructed from two complementary sources, LAION and RealQA, which provide English/Chinese content distributions.
>
> **Uniqueness**. The main distinction of UGC-Edit is focusing specifically on authentic user-captured photos for autonomous photo editing. Existing datasets (AVA, AADB, and PARA) are designed for general aesthetics assessment, whereas our target is everyday user photos.
>
> > 5. The reward model is both trained and evaluated on the same dataset.
>
> We clarify that the reward model is trained on UGC-Edit, but evaluated externally on PARA, which does not overlap with the training data (**Appendix B**). The results on PARA suggest that the model generalizes beyond its training distribution rather than simply memorizing UGC-Edit. More importantly, our main conclusions do not rely on the reward model alone, and the gains are also supported by independent human preference results.
>
> > 6. Efficiency is also a concern.
>
> **We report the Latency and speed–quality trade-offs in Appendix F**. The default configuration is quality-oriented. However, the appendix also demonstrates substantially faster operating points. For example, reducing the search budget from 20 to 5 simulations decreases runtime from approximately 470s to 60s, with only a modest degradation in quality (BRISQUE 0.6217 → 0.6292).

---

> > ### Author Rebuttal · Reviewer_mFre · 2026-04-04
> >
> > Thank you for the detailed rebuttal. The clarifications regarding the planning formulation and the added NanoBanana2.0 / Grok comparisons are helpful and improve the positioning of the system.
> >
> > However, one of my main concerns remains only partially addressed: it is still somewhat unclear what new scientific insight emerges beyond demonstrating that multi-step planning improves editing quality over single-step pipelines. The argument that long-horizon exploration outperforms reactive editing is plausible and empirically supported, but the paper would benefit from a clearer articulation of what research questions this framing enables beyond system-level performance improvements. As written, the contribution still reads primarily as an integration and coordination strategy rather than a conceptual advance.
> >
> > Regarding the added executor baselines, I appreciate that NanoBanana2.0 and Grok were evaluated in supplementary materials. However, the comparison is currently qualitative. Since these tools are also components within the broader system pipeline, including even a small quantitative comparison under the same evaluation protocol would make it easier to assess how much improvement comes from planning versus executor choice.
> >
> > Thank you for explaining points 3 and 5, no questions remain there. For point number 4 can you explain how your dataset is different from RealEdit or MultiRef.
> >
> > Overall, the rebuttal improves confidence in the implementation and evaluation details, but the questions about conceptual novelty remain partially open.

---

> > > ### Author Response · Authors · 2026-04-08
> > >
> > > Thank you for the constructive feedback. We thank the reviewer for the continued engagement and for acknowledging the improvements of our work. We address the remaining concerns below.
> > >
> > >
> > > 1. Novelty
> > >
> > > We have two points for further clarification. **First**, beyond improving quality, our agentic editing helps users avoid designing complex prompts. Users often do not know exactly what parameters to give since they provide only vague input like "make it better". Our agentic system autonomously decomposes the intent, plans the sequence of operations, and iterates, which produces high-quality results and relieves the user of prompt-engineering burden. **Second**, simply integrating existing tools is insufficient as they lack coordination and fail under ambiguous instructions. Our system makes them work together effectively.
> > >
> > > 2. Quantitative Comparison of Executors
> > >
> > > We have conducted a quantitative evaluation using the same protocol as our main experiments on a randomly sampled subset of 100 images, as shown below.  The results show that while executors provide a good quality, our method provides a boost based on the underlying executor.
> > >
> > > ### Comparison of Average Metrics Across Models
> > >
> > > | Model                          | CLIP score ↑ | Laion Reward ↑ | ImageReward score ↑ | UGC score ↑ |
> > > |--------------------------------|-------------|----------------|---------------------|-------------|
> > > | Grok-imagine                   | **0.6056**  | 0.5129         | **0.4322**          | 4.310       |
> > > | Gemini-3.1-flash-image-preview | 0.6023      | _0.5142_       | 0.4148              | _4.349_     |
> > > | Ours                           | _0.6025_    | **0.5153**     | _0.4314_            | **4.368**   |
> > >
> > >
> > >
> > > 3. Dataset comparison with RealEdit and MultiRef
> > >
> > > Our UGC-Edit dataset differs from RealEdit and MultiRef in two key aspects. **First**, RealEdit focuses on image editing training, and MultiRef on controllable image generation, but neither supports evaluation. In contrast, our dataset is designed as a scoring dataset for aesthetic quality, enabling the training of a reward model, which is essential for closing the loop in agent-based UGC image editing. **Second**, our dataset is specifically collected for user-generated content (UGC) for aesthetic evaluation, which is not covered in prior datasets. We will clarify differences with these two datasets more explicitly in the paper.

---

### Official Review · Reviewer_VkRR · 2026-03-10

**Soundness:** 3
**Presentation:** 3
**Significance:** 3
**Originality:** 3
**Overall Recommendation:** 5
**Confidence:** 3

**Summary:**

This paper presents PhotoAgent, an autonomous image editing system that frames photo editing as a long-horizon sequential decision-making problem. The system consists of four components: a VLM-based perceiver that generates candidate editing actions, an MCTS-based planner that explores the action space, a tool-based executor, and a multi-metric evaluator. To address the lack of UGC-specific evaluation data, the authors also introduce UGC-Edit, a dataset of ~7,000 real user photos annotated with aesthetic scores, and train a reward model on it using GRPO. Experiments on a self-constructed benchmark of 1,017 images show PhotoAgent outperforming a range of single-step and agent-based baselines, particularly on the BRISQUE metric.

**Compliance With Llm Reviewing Policy:**

Affirmed.

**Final Justification:**

After Authors' rebutta, my concers have been solved. I will raise my score.

**Key Questions For Authors:**

Please see weakness.

**Limitations:**

Yes, in Appendix G.

**Strengths And Weaknesses:**

Strength

1. The application of MCTS to autonomous photo editing is a natural and well-motivated extension from game-playing and text-generation domains.

2. The two-stage filtering pipeline UGC-Edit of  (VLM categorization followed by manual verification) is a reasonable approach to quality control.

3. The computational cost analysis (Appendix F) is unusually transparent.

Weakness

1. The overall quantitative gains are modest and raise reproducibility concerns. On several key metrics (ImageReward, Laion-Reward), PhotoAgent is only marginally better than GPT-4o or closed-loop ReAct.

2. The MCTS state space design is underspecified. Each "state" is a full image, yet the action space is a natural-language editing instruction.

3. Overall quantitative gains are modest and raise reproducibility concerns. On several key metrics (ImageReward, Laion-Reward), PhotoAgent is only marginally better than GPT-4o or closed-loop ReAct. Meanwhile, the BRISQUE score—where PhotoAgent's advantage is most pronounced—primarily reflects signal-level distortion rather than subjective aesthetic quality.

---

> ### Author Rebuttal · Authors · 2026-03-31
>
> We sincerely thank the reviewer for the positive assessment of our work and insightful comments. We hope our response below can address the reviewer's concerns.
>
> > 1. The overall quantitative gains are modest and raise reproducibility concerns. On several key metrics (ImageReward, Laion-Reward), PhotoAgent is only marginally better than GPT-4o or closed-loop ReAct.
>
>
> (1)	PhotoAgent is the only method that ranks top-2 on every metric.
>
>
> - Compare to GPT-4o: Our method is leading with significant margin in fidelity score, such as BRISQUE (improved ~0.10) and CLIP Similarity (+0.0239), with a marginal drop in ImageReward (-0.0036) and UGC Score (-0.034).
>
> - Compared with ReAct (the strongest agent baseline), PhotoAgent leads on all five metrics, with especially large margins on CLIP Similarity (+0.0227), BRISQUE (-0.0205), and UGC Score (+0.918). No other method achieves this balance between aesthetic quality and content preservation.
>
> (2) The user study offers complementary and more direct evidence. PhotoAgent obtains a substantial margin that reflects real perceptual preference and validates the practical value of our approach beyond automatic metrics.
>
> (3) Regarding reproducibility, we will release our code, trained reward model, UGC-Edit dataset, and evaluation benchmark upon acceptance.
>
>
> > 2. The MCTS state space design is underspecified. Each "state" is a full image, yet the action space is a natural-language editing instruction.
>
> We clarify the state and action space as follows.
>
> **State.** Each state \(s_k\) is indeed an image, representing the current editing result with k-th action. States are generated on-the-fly as the tree is expanded through action execution.
>
> **Action space.** The action space at each node consists of the actions generated by the Perceiver. Specifically, the Perceiver generates exactly \(K\) candidate editing instructions for each state, so planner selecting an action is equivalent to choosing from \(\{1, 2, ..., K\}\). These instructions are expressed in natural language (NL) because the downstream executors consume NL inputs, and the MCTS planner treats them purely as discrete choices.
>
> We will clarify these explicitly in the revised manuscript.
>
> > 3. Overall quantitative gains are modest and raise reproducibility concerns. On several key metrics (ImageReward, Laion-Reward), PhotoAgent is only marginally better than GPT-4o or closed-loop ReAct. Meanwhile, the BRISQUE score—where PhotoAgent's advantage is most pronounced—primarily reflects signal-level distortion rather than subjective aesthetic quality.
>
> We thank the reviewer for this comment. We respectfully note that the five reported metrics are designed to be complementary, each capturing a distinct quality dimension: content preservation (CLIP Similarity), human preference (ImageReward), naturalness (BRISQUE), aesthetic preference (Laion-Reward), and UGC-specific quality (UGC Score).
>
> PhotoAgent is the only method that simultaneously ranks in the top-2 across all five metrics. No other method achieves this holistic balance. GPT-4o scores well on preference-based metrics but degrades on BRISQUE and CLIP similarity (naturalness and content preservation). Agent baselines perform inconsistently across metrics. We believe this consistent, balanced superiority is more meaningful than a large margin on a single metric.

---

> > ### Author Rebuttal · Reviewer_VkRR · 2026-04-03
> >
> > Thanks for Authors' rebuttal. My concers have been solved.

---

> > > ### Author Response · Authors · 2026-04-08
> > >
> > > Thanks for acknowledging our efforts and work! We will incorporate your feedback into the revision.

---

### Official Review · Reviewer_kjVY · 2026-03-11

**Soundness:** 3
**Presentation:** 3
**Significance:** 3
**Originality:** 3
**Overall Recommendation:** 5
**Confidence:** 2

**Summary:**

This paper presents PhotoAgent, an autonomous image editing system that addresses the limitations of current instruction-based editing models that require detailed, expert-level prompts. The system uses a closed-loop architecture consisting of four components. In addition, the authors construct the UGC-Edit dataset, containing 7K+ images of user-generated content with editing intent, and train a Qwen-2.5-VL-based reward model via GRPO to align with human aesthetic preferences. The experiments show that PhotoAgent outperforms single-step baselines and other agent-based methods across semantic alignment and aesthetic quality metrics.

**Compliance With Llm Reviewing Policy:**

Affirmed.

**Final Justification:**

The authors' response has adequately addressed my concerns. I recognize the value of this work and maintain my recommendation to accept the paper.

**Key Questions For Authors:**

1. Can the proposed method adjust itself autonomously based on the input without a prompt?
2. Will the UGC-Edit Dataset be released?

**Limitations:**

1. The requirement for higher computing resources and latency is a factor that needs to be taken into consideration.
2. A User study with too few participants introduces a high degree of randomness. In addition, are the participants of the user study diverse? In many cases, people's aesthetic perceptions are rich.

**Strengths And Weaknesses:**

### Strengths

1. The proposed UGC-Edit dataset addresses an important gap in existing benchmarks.
2. The motivation of the article is clear and innovative.
3. The results of SoTA are achieved in several comparative experiments.

### Weaknesses

See Questions and Limitations.

---

> ### Author Rebuttal · Authors · 2026-03-31
>
> We sincerely thank the reviewer for the positive assessment of our work and insightful comments. We hope our response below can address the reviewer's concerns.
>
> > 1. Can the proposed method adjust itself autonomously based on the input without a prompt?
>
> Yes, the proposed method can operate autonomously without requiring an explicit user prompt. We clarify this capability from three aspects:
>
> (1) The method is explicitly designed to support both fully autonomous editing and user-guided editing within a unified framework. This dual-mode design allows the system to flexibly adapt to different usage scenarios.
>
> (2) In the autonomous editing mode (without user prompt), the perceiver leverages a predefined system-level prompt (Appendix H) to analyze the input image and infer appropriate editing intentions and action sequences.
>
> (3) In the user-guided editing mode (with user prompt), the user-provided prompt is incorporated as high-level guidance. The model interprets this guidance to plan a sequence of intermediate actions, which are then executed in a step-by-step manner. In this case, the prompt influences the overall editing direction.
>
> > 2. Will the UGC-Edit Dataset be released?
>
> Yes, the UGC-Edit dataset will be released for public use.
>
> > 3. The requirement for higher computing resources and latency is a factor that needs to be taken into consideration.
>
> We thank the reviewer for this thoughtful and constructive comment. We report the current latency as well as the speed–quality trade-off under different search budgets in Appendix F. The default configuration is quality-oriented. However, the appendix also demonstrates substantially faster operating points. For example, reducing the search budget from 20 to 5 simulations decreases runtime from approximately 470s to 60s, with only a modest degradation in quality (BRISQUE 0.6217 → 0.6292).
>
> We have also discussed several directions to further reduce cost and latency, including reducing the number of editing steps, adopting lighter backbone models, and applying model compression or caching strategies. We believe that efficiency optimization can be further improved in future iterations.
>
> > 4. A User study with too few participants introduces a high degree of randomness. In addition, are the participants of the user study diverse? In many cases, people's aesthetic perceptions are rich.
>
> We thank the reviewer for this thoughtful comment. We thank the reviewer for this important comment. While the number of participants is moderate, our study collects **hundreds of comparisons under a controlled protocol**, which substantially reduces randomness and provides a reliable estimate of human preference. Such evaluation is widely used for its higher sensitivity and consistency compared to absolute scoring.
>
> Specifically, we recruit participants with varying levels of photography and image-editing experience, **ranging from casual smartphone photographers to professional photographers** to ensure diversity. Moreover, all comparisons follow the same criteria and setup. Participants are instructed to: "Select the image with the best editing result. The criteria are: while preserving the original photo content as much as possible, the edited image should be more aesthetically pleasing overall and something you would be more willing to share.'' We will further expand the scale and diversity of the study in future work.

---

> > ### Author Rebuttal · Reviewer_kjVY · 2026-04-01
> >
> > The author's response has addressed my concerns. For agent-related work, indeed, it is not superior to the end-to-end model in terms of efficiency, but for me, I agree that this is a trade-off between effectiveness and efficiency and not a problem unique to PhotoAgent, and the author's response is also convincing.
> >
> > Besides, I think the UGC-Edit dataset presented in this article is a contribution to the community, and I recognize the value of this paper for the work of agents, which cannot be viewed from the perspective of end-to-end models. So I keep my score, thanks to the author.

---

> > > ### Author Response · Authors · 2026-04-08
> > >
> > > We sincerely thank the reviewer for the insightful comments which helped us refine our paper. We are very pleased to hear that the reviewer acknowledges the significance of our findings.

---

### Decision · Program_Chairs · 2026-04-30

**Decision:**

Accept (spotlight)

**Comment:**

The reviewer consensus is positive, and I strongly recommend accepting this paper. PhotoAgent offers a refreshing and highly practical approach to autonomous image editing by framing it as a long-horizon, MCTS-based decision process rather than a standard single-step generation task. Coupled with the release of the UGC-Edit dataset, the authors deliver a substantial, systemic contribution to the field. The committee specifically highlighted the following strengths:

Architectural Innovation: By moving away from,single-step instruction models, the closed-loop design is much better equipped to handle the kind of ambiguous, open-ended intent we actually see in practice.

High-Value Dataset: The UGC-Edit dataset is a major highlight for the community. Finding reliable benchmarks for authentic user-generated content is an ongoing bottleneck, and this dataset fills a critical gap for evaluating genuine aesthetic quality—allowing the system to demonstrate top-tier performance across a balanced suite of metrics.